# Dietary Glycemic Index and Load and the Risk of Type 2 Diabetes: Assessment of Causal Relations

**DOI:** 10.3390/nu11061436

**Published:** 2019-06-25

**Authors:** Geoffrey Livesey, Richard Taylor, Helen F. Livesey, Anette E. Buyken, David J. A. Jenkins, Livia S. A. Augustin, John L. Sievenpiper, Alan W. Barclay, Simin Liu, Thomas M. S. Wolever, Walter C. Willett, Furio Brighenti, Jordi Salas-Salvadó, Inger Björck, Salwa W. Rizkalla, Gabriele Riccardi, Carlo La Vecchia, Antonio Ceriello, Antonia Trichopoulou, Andrea Poli, Arne Astrup, Cyril W. C. Kendall, Marie-Ann Ha, Sara Baer-Sinnott, Jennie C. Brand-Miller

**Affiliations:** 1Independent Nutrition Logic Ltd, 21 Bellrope Lane, Wymondham NR180QX, UK; richard.mi.taylor1@gmail.com (R.T.); helenlivesey@inlogic.co.uk (H.F.L.); 2Institute of Nutrition, Consumption and Health, Faculty of Natural Sciences, Paderborn University, 33098 Paderborn, Germany; anette.buyken@uni-paderborn.de; 3Departments of Nutritional Science and Medicine, Faculty of Medicine, University of Toronto, Toronto, ON M5S 1A8, Canada; david.jenkins@utoronto.ca (D.J.A.J.); john.sievenpiper@alumni.utoronto.ca (J.L.S.); thomas.wolever@utoronto.ca (T.M.S.W.); cyril.kendall@utoronto.ca (C.W.C.K.); 4Clinical Nutrition and Risk Factor Modification Centre, St. Michael’s Hospital, Toronto, ON M5C 2T2, Canada; livia.augustin@utoronto.ca; 5Division of Endocrinology and Metabolism, Department of Medicine, St. Michael’s Hospital, Toronto, ON M5C 2T2, Canada; 6Li Ka Shing Knowledge Institute, St. Michael’s Hospital, Toronto, ON M5C 2T2, Canada; 7Epidemiology, Istituto Nazionale Tumori IRCCS “Fondazione G. Pascale”, 80131 Napoli, Italy; 8Glycemic Index Foundation, 26 Arundel St, Glebe, Sydney NSW 2037, Australia; alanb@gifoundation.org.au; 9Department of Epidemiology and Medicine, Brown University, Providence, RI 02912, USA; simin_liu@brown.edu; 10Departments of Nutrition and Epidemiology, Harvard T. H. Chan School of Public Health and Harvard Medical School, Boston, MA 02115, USA; wwillett@hsph.harvard.edu; 11Department of Food and Drug, University of Parma, 43120 Parma, Italy; furio.brighenti@unipr.it; 12Human Nutrition Unit, Department of Biochemistry and Biotechnology, Faculty of Medicine and Health Sciences, Institut d’Investigació Sanitària Pere Virgili (IISPV), Rovira i Virgili University, 43201 Reus, Spain; jordi.salas@urv.cat; 13Fisiopatología de la Obesidad y Nutrición (CIBEROBN), Instituto de Salud Carlos III, 27400 Madrid, Spain; 14Retired from Food for Health Science Centre, Antidiabetic Food Centre, Lund University, S-221 00 Lund, Sweden; inger@innovafood.se; 15Institute of Cardiometabolism and Nutrition, ICAN, Pitié Salpêtrière Hospital, F75013 Paris, France; salwa.rizkalla3@gmail.com; 16Department of Clinical Medicine and Surgery, Federico II University, 80147 Naples, Italy; riccardi@unina.it; 17Department of Clinical Sciences and Community Health, Università degli Studi di Milano, 201330 Milan, Italy; carlo.lavecchia@unimi.it; 18IRCCS MultiMedica, Diabetes Department, Sesto San Giovanni, 20099 Milan, Italy; ACERIELL@clinic.cat; 19Hellenic Health Foundation, Alexandroupoleos 23, 11527 Athens, Greece; atrichopoulou@hhf-greece.gr; 20Nutrition Foundation of Italy, Viale Tunisia 38, I-20124 Milan, Italy; poli@nutrition-foundation.it; 21Department of Nutrition, Exercise and Sports (NEXS) Faculty of Science, University of Copenhagen, 2200 Copenhagen, Denmark; ast@nexs.ku.dk; 22College of Pharmacy and Nutrition, University of Saskatchewan, Saskatoon, SK S7N 5B5, Canada; 23Spinney Nutrition, Shirwell, Barnstaple, Devon EX31 4JR, UK; nutrition@thespinney.co.uk; 24Oldways, Boston, MA 02116, USA; sara@oldwayspt.org; 25Charles Perkins Centre and School of Life and Environmental Sciences, University of Sydney, Sydney NSW 2006, Australia; jennie.brandmiller@sydney.edu.au

**Keywords:** causation, diabetes, glycemic index, glycemic load, dietary fiber, alcohol, cohort studies, epidemiology, meta-analysis, public health

## Abstract

While dietary factors are important modifiable risk factors for type 2 diabetes (T2D), the causal role of carbohydrate quality in nutrition remains controversial. Dietary glycemic index (GI) and glycemic load (GL) have been examined in relation to the risk of T2D in multiple prospective cohort studies. Previous meta-analyses indicate significant relations but consideration of causality has been minimal. Here, the results of our recent meta-analyses of prospective cohort studies of 4 to 26-y follow-up are interpreted in the context of the nine Bradford-Hill criteria for causality, that is: (1) Strength of Association, (2) Consistency, (3) Specificity, (4) Temporality, (5) Biological Gradient, (6) Plausibility, (7) Experimental evidence, (8) Analogy, and (9) Coherence. These criteria necessitated referral to a body of literature wider than prospective cohort studies alone, especially in criteria 6 to 9. In this analysis, all nine of the Hill’s criteria were met for GI and GL indicating that we can be confident of a role for GI and GL as causal factors contributing to incident T2D. In addition, neither dietary fiber nor cereal fiber nor wholegrain were found to be reliable or effective surrogate measures of GI or GL. Finally, our cost–benefit analysis suggests food and nutrition advice favors lower GI or GL and would produce significant potential cost savings in national healthcare budgets. The high confidence in causal associations for incident T2D is sufficient to consider inclusion of GI and GL in food and nutrient-based recommendations.

## 1. Introduction

Preventing type 2 diabetes (T2D) is a major goal of the World Health Organization, the International Diabetes Federation and many national organizations around the globe. There is scientific consensus that modifiable lifestyle choices influence a person’s risk of developing T2D. Known risk factors include energy intake, dietary fat, refined carbohydrate, refined grains, alcohol, dietary fiber, whole grains, overweight and obesity, physical activity and smoking [1]. It is possible that consuming diets in which the carbohydrate is too high in glycemic index (GI) or load (GL) may also constitute an increased risk, but the case for causality and the appropriate recommendations to include in public health guidelines are less certain. In the present context, GI reflects the glycemic impact of the carbohydrates in a diet (irrespective of quantity) while GL represents the overall glycemic impact taking into account both quantity and quality of carbohydrate.

While it is established that lowering of fasting hyperglycemia can reduce both the risk of insulin resistance and β-cell dysfunction [2,3,4], it is questioned whether decreasing postprandial glycemia has similar effects. Moreover, it is controversial whether choosing lower GI or GL foods can reduce day-long postprandial glycemia. Thus, the question of cause-and-effect relationships between T2D and GI and GL remains. Here, we examine causality according to Bradford-Hill criteria [5], which is a recognized tool for interpreting both observational and interventional evidence, and the case for making dietary attributes (e.g., GI and GL) part of dietary recommendations [6,7,8,9]. We briefly include a cost–benefit consideration because this is also relevant to the justification for advising on a change of dietary habits within a population [10]. Critical to this inquiry is our updated systematic review with meta-analyses [11].

## 2. Methods

### 2.1. Bradford Hill-Criteria

The Bradford-Hill criteria, otherwise known as Hill’s criteria for causation, are a group of guidelines that can be useful for providing evidence of a causal relationship between a putative cause and an effect, and were established by the English epidemiologist Sir Austin Bradford Hill in 1965 [5]. The criteria fall under headings or viewpoints: strength of association (from meta-analyses where possible), consistency of association, specificity, temporality, biological gradient (dose-dependency), plausibility, experimental evidence, analogy, and coherence. These criteria with original definitions are shown in Table 1 alongside the definitions as used in the assessment of causation in the present analysis. For simplicity, a score of 1 was given to each criterion fulfilled. Confidence in causation “the most likely [probable] interpretation“ as stated by Hill [6] was considered to increase as the sum of criteria fulfilled increased from 1 to 9; scores of 1 to 3 are considered low, 4 to 6 moderate, and 7 to 9 high.

### 2.2. Biological Gradient (Dose-Response) Analysis

Biological gradient (dose-response) meta-analyses for both GI and GL were undertaken using the generalized least-squares method for trend estimation of the dose-response data [12] (Stata software version 11.2 SE, 2009; StataCorp LP) using the pool first option [13], which is a one-step procedure. These were termed local dose-response analyses rather than the global dose-response analyses in [11] and were additional analyses to those undertaken in [11]. They differ in that local dose response meta-analyses have maximally overlapping exposures such that the combined study result gives the weighted mean RR within jurisdictions rather than across jurisdictions (areas of land having particular forms of law). Eligible prospective cohort studies were those that provided mean and 95% confidence limits for RR/HR/OR and the corresponding doses (exposures) for GI and GL obtained using valid dietary instruments defined as having a validity correlation for carbohydrate > 0.55 [11,14]. Data on GI, GL, risk ratios or risk relations, extracted from the original published prospective cohort studies by quantile and dose are given in the Supplemental Material files in [11].

## 3. Results

Table 2 summarizes our assessment in accordance with our interpretation of Bradford Hill criteria for causation of T2D by GI and GL (Table 1) and is placed immediately after the results.

### 3.1. Strength of Association

In addition to our updated meta-analyses reported in [11], we found 6 prior meta-analyses of the T2D-GI and T2D-GL risk ratios (RR as point estimate) or risk relations (RR as rate estimates) that were based on prospective cohort studies published in peer reviewed journals. All reported finding statistically significant links between T2D and GI [17,18,19,20,21,22] and T2D and GL [17,18,19,20,21] with one exception—[22] found no link at all for the T2D-GL risk ratio (Table SUP3:1 in [22]), while finding a T2D-GI risk ratio that increased by 12% with higher GI (without information on the corresponding range of GI intakes) (Table S1 in [22]).

Errors are evident in all these meta-analyses, of types described in [14], however the errors appear to be many in [22] which we focus on as the most recent high impact paper assessing carbohydrate quality: (1) Combining studies with different definitions for the range of exposure (tertiles, quartiles, and quintiles) in extreme quantile meta-analysis (EQM) without evidence of account for these differences. (2) Within definitions of exposure not accounting for different ranges of exposure. (3) Ignoring intermediate quantiles, so not making maximum use of the available data. (4) Not succeeding in selecting studies for the longest duration of follow-up. (5) For GL not adjusting for the different levels of energy intake among studies. (6) Ignoring studies that provided risk relations rather than risk ratios by quantiles. (7) Lack of attention to the totality of evidence since there was no obvious list explaining why some studies were excluded. (8) Lack of clarity in model result selection from the original studies. Typically fully adjusted model results are used but not all studies included in their meta-analyses published their fully adjusted models. (9) Not accounting for studies using inadequately validated dietary instruments, which affects comparability of studies and is reason for excluding them [23]. (These can be set aside to be used when a meta-analytical model is found that enables their inclusion.) (10) Not seeking to identify outlying studies; this is problematic because observational studies frequently yield spurious results [24], and which in observational studies are not usually better captured by selecting a non-normal distribution of observations. (These studies also can be set aside to be used when a meta-analytical models is found that includes them.) (11) When attempting global dose-response meta-analysis, incorrect data entry resulting in: (a) a wide range of null outputs where there are no data inputs and apparently. (b) individual study referents planted on the null rather than on the forecasted study position (these were most evident for the T2D-GI risk relation in their Figure 3C in [22]). (12) No inclusion or exclusion criteria were made clearly evident for the retrieved publications. (13) Limited or no evidence of attempt to explain heterogeneity other than dropping individual studied. A broader assessment of sources of heterogeneity has been regarded as most important for observational studies [24]. These problems are not unique to [22] (except possibly 11) and are present to a lesser extent in other meta-analyses, especially those undertaking EQM.

Thus, including the various errors, published meta-analyses have reported combined mean RR values that ranged from 1.12 to 1.45 for GI, and 1.12 to 1.45 also for GL (i.e., incremental risks of from 12% to 45% over approximately 4 to 5 categories/quantiles of exposure). The reports included 6 to 24 original studies in meta-analyses. When assessing the T2D-GI risk ratios, the highest value arose from a meta-analysis that had excluded studies using food frequency questionnaires with inadequate validity correlations for nutrients in general [18]. When assessing the T2D-GL risk relation, the highest RR value was obtained if no studies were excluded on this ground, but adjustment was made for the validity correlation coefficient and rounding the risk relation to per 100 g GL/d in 2000 kcal [17]. Not taking account of the validity correlation can lead to serious underestimation of disease-nutrient RR values [11,14,17,18], which can leave health professionals and patients ill-informed to their detriment. It could lead also to extreme beliefs that epidemiology is worthless, which would be falsely based. The validity correlation is the correlation between the intakes of a nutrient or type of food estimated from a dietary instrument (usually a food frequency questionnaire) and the intake determined using food records among a sample of the population to be studied. Optimally these correlations are energy adjusted and deattenuated [25] and are sufficiently large to validate the questionnaire when judged against a predefined value [26]. If one uses a dietary instrument to assess the nutrient content of diets from foods people eat and this content associates (correlates) poorly with measured dietary nutrient contents using robust food records, one can only expect an erroneously low association with a disease outcome, too.

For a harmful RR, a value > 1.20 with lower 95% confidence limit > 1.10 (irrespective of the level of inconsistency, I^2^) across the 10th to 90th percentiles of intakes (approximately from the median of the 1^st^ to the median of the 5th quintile) can be considered to be clinically or biologically important in dietary studies for nutrient exposure posing a risk to public health [11,14,15]. The spread of RR values from the variously published meta-analyses (1.12 to 1.45, see above) fell from above to below 1.20. Because of this and because there were multiple problems in the preparation of observational data in almost all those meta-analyses (as noted above), it was important first to determine the strengths of the T2D-GI and GL risk relations. Thus we updated the first meta-analyses that accounted for the validity correlation [18] with more recent studies but applied dose-response meta-analyses (DRM) rather than EQM. Where possible, we also addressed the hypothesized sources of heterogeneity. These were considered in detail in our most recent systematic review and dose-response meta-analyses [11].

Using eligible, valid prospective cohort studies (those applying valid dietary instruments with correlation coefficients > 0.55 for carbohydrate), the dose-response incident T2D-GI and GL risk relations, the primary outcomes, were: 1.27 (1.15–1.40) (*P* < 0.001, *n* = 10) per 10 units GI and 1.26 (1.15–1.37) (*P* < 0.001, *n* = 15) per 80 g GL in 2000 kcal (8400 kJ) diet (Table 7 row 2 for GI and GL in [11]). The 10 units GI and 80 g/d GL in 2000 kcal diet corresponded to the range of exposures approximately across the 10th to the 90th percentile of typical population intakes (for the included studies).

Across the wider global (worldwide) ranges of intakes, dose-response meta-analyses yielded increments in RR of 1.87 (1.56–2.25) across 35 units GI and 1.89 (1.66–2.16) across 235 g/d GL in 2000 kcal (8400 kJ) diet. We say “across” rather than “per”, because the dose-responses were not log-linear over the global range of intakes, with strongest risks across the lower range of intakes (Sections 3.2.17 and 3.3.19 in [11]).

Over all we judged the T2D-GI and GL risk relations for the primary outcomes to be more likely underestimated than overestimated. Indeed, higher RR values were obtained when selecting for studies using clinical ascertainment of T2D and study-level adjustment for family history of diabetes (FHD) (Table 7 in [11]) or adjusting for the validity correlation and family history of diabetes as covariates (Table 8 in [11]).


*Criterion conclusion:*
Critical meta-analyses of prospective cohort studies show both the T2D-GI and the T2D-GL relations are sufficiently strong to consider action in favor of public health.


### 3.2. Consistency of Association

Sources of inconsistency (I^2^) among eligible prospective cohort studies on the T2D-GI and GL risk relations were uncovered. Almost all inconsistency (or heterogeneity) was explainable, (I^2^, 0 to 1%) (Tables 7 and 8 in [11]). Meanwhile relevant funnel plots were symmetrical implying absence of small-study effects, publication bias or selection bias (Sections 3.2.4 and 3.3.6 in [11]).

Note that the criteria of interest for public health in Table 1 (Strength) make no specific requirement about the level of inconsistency (I^2^) of results among studies. This is because meeting the criteria would not be possible if inconsistency was too high. This is useful because I^2^ is an imprecise measure [27,28]. Indeed, I^2^ can be zero when studies are imprecise and can be high when a risk relation is large.

#### 3.2.1. The T2D-GI Risk Relation

Eighty-percent of studies using a dietary instrument with validity correlation > 0.55 gave results in the same direction, which was a higher risk associated with higher GI (Figure 3 in [11]). Those not doing so had upper 95% confidence limits of 1.35 and 1.74 (Figure 3 in [11]), and therefore were not statistically inconsistent with the higher risk at higher dietary GI.

Use of dietary instruments with a low validity correlation for carbohydrate (≤0.55) significantly underestimated the T2D-GI risk relation by 5 to 6 fold compared with those with validity correlations > 0.55 (*P* = 0.004) (Section 3.2.2 in [11]). However, studies using dietary instruments with validity correlations ≤ 0.55 were includable when using the validity correlation as a covariate (*P* < 0.001) together with study-level adjustment for family history of diabetes (*P* < 0.001, *n* = 15 studies, Section 3.2.13 in [11]). In this instance there was essentially no inconsistency (I^2^ = 1%) and the risk relation remained sufficiently high to meet the public health acceptability criteria (1.29 (1.23–1.35) (*n* = 15 studies, *P* < 0.001) (Table 8, row 3 for GI in [11]).

Among studies with validity correlations > 0.55 there was no significant evidence of missing studies or publication or selection bias that could suggest overestimation of RR or underestimation of I^2^ (Section 3.2.4 in [11]).

#### 3.2.2. T2D-GL Relation

Ninety-three percent of studies using a dietary instrument with a validity correlation > 0.55 for carbohydrate gave results in the same direction, which was a higher risk at higher GL (Figure 7, in [11]). Those studies not doing so had upper 95% confidence limits of 1.53 and 1.87 and therefore were not statistically inconsistent with the higher RR at higher dietary GL.

Use of dietary instruments with a low validity correlation for carbohydrate (≤0.55) underestimated the T2D-GL risk relation by 3-fold compared to studies with validity correlations > 0.55 (Table 4, rows 1 vs. 2 in [11]). This difference did not reach statistical significance (*P* = 0.078, *n* = 21 studies) (Section 3.3.2 in [11]) but was in the expected direction. Moreover, significance was reached when CORR was used as a covariate across all studies (*P* < 0.006, *n* = 22 studies without excluding studies with validity correlations ≤ 0.55 or outliers) (Section 3.3.16, para. 3 in [11]). Further the validity correlation also retained significance for studies with this correlation > 0.55 (*P* < 0.001, *n* = 15 studies with the study of Simila et al. outlying, *P* = 0.010) (Table 4 row 4 footnote *g* in [11]). In this instance the inconsistency statistic (I^2^) was reduced to 0%. Again the risk relation was sufficiently high (RR = 1.36 (1.26–1.48) per 80 g GL in 2000kcal diet (Table 4, row 4)) to meet the public health acceptability criteria (RR > 1.20 and L95%CL > 1.10).

As for GI, among studies on GL with validity correlations > 0.55 there was no significant evidence of small-study effects, e.g., publication bias or study selection bias, that could suggest overestimation of RR or underestimation of the inconsistence statistic (I^2^%) (Section 3.3.6 in [11]).

#### 3.2.3. Significant Sources of Inconsistency (or heterogeneity)

*The validity correlation:* Several source of inconsistency (heterogeneity) were encountered, the most important of which was the validity correlation coefficient for carbohydrate. Placing this in some historical context, the errors in measurement of dietary intake have been reviewed [25] with important subsequent contributions on the validity of food frequency questionnaires (FFQ) [26,29], which now find common use in epidemiology. Brunner et al. [26] found that for most nutrients, a validity correlation of 0.5 was sufficient to rank most nutrient intakes when using an FFQ but that specific values can arise; for example 0.6 for Southgate dietary fibre and 0.8 for alcohol. Barclay et al. used a validity correlation of >0.50 for nutrients in general to select for reliable studies of the T2D-GI and GL risk ratios in meta-analysis [26], while [14] used >0.55 in-meta analysis of the CHD-GI and GL risk relations. Livesey et al. [17] had found there was no specific cut-point for the T2D-GL risk relation, rather the validity correlation was approximately continuously higher with higher T2D-GL risk relations and this was also evident when the validity correlation was >0.55 (Figure 7 in [11]). Thus, variation in study-level RR values could be adjusted for the validity correlation both when withdrawing studies with validity correlations ≤ 0.55 [11] and when not withdrawing these studies [17] but using the validity correlation as a covariate. At the outset we handled this source of inconsistency in results by withdrawing studies with CORR ≤ 0.55 (for our primary outcomes) only and re-include in secondary outcomes should an analytical model become evident for which CORR was a significant covariate. CORR was a significant covariate and source of inconsistency in the T2D-GL risk relations [11,17] and became a significant covariate and source of inconsistency in the T2D-GI risk relation when simultaneously adjusting for family history of diabetes as a covariate (Section 3.2.13 in [11]).

*Clinical ascertainment of diabetes.* In keeping with the earliest published meta-analyses of the T2D-GI and GL risk ratios [18], greater risk relations were found in our update using dose-response meta-analyses among studies using clinical reported T2D than self-reported T2D (Table 7 rows 3 for GI and row 3 for GL in [11]).

*Family history of diabetes.* Study- level adjustments for FHD as a covariate in meta-analysis was a significant source of heterogeneity (*P* < 0.001) independently of the validity correlation for carbohydrate. FHD may also have been a source of inconsistency for the T2D-GL risk relation independently of the validity correlation, but did not reached statistical significance (*P* = 0.14, Section 3.3.16 in [11]) though remains plausible because of the significance for the T2D-GI risk relation while GI is a component of GL.

*Alcohol.* Both the T2D-GI and GL risk relations were attenuated in studies in which the mean population alcohol consumption was moderately high (>15 g/d). Likewise, a higher than average alcohol consumption completely attenuated the coronary heart disease-GI and GL risk relations [14]. This attenuation may be model dependent and for T2D (and CHD [14]) was found when the validity correlation was also a covariate. This attenuation was first identified within a prospective cohort study for GL though not for GI [30]. The observations within this [30] study were weaker than we found among studies for T2D [1] and CHD [14], which might be due to the within-study observations applying to women only whereas the among-study observations were in men and women combined. A possibility exists that a sex difference in the size of the T2D-GI and GL risk relations may be due to a sex difference in alcohol consumption [1]. Attenuation by alcohol consumption may explain why observations from the Healthy Professionals Follow-up Study [19] and 3 other studies [31,32,33] showed lower than expected associations between T2D and GL. Epidemiological observations do not identify a mechanism of action, which could be many [30]. Nevertheless, our evidence points toward higher T2D (and CHD [14])-GI and GL risk relations for alcohol abstainers than indicated by the primary outcome risk relations.

Sex of participants. There was no evidence of a sex-specific difference in the size of the dose-response T2D-GI risk relation (Table 1 in [11]), though one was evident for the T2D-GL risk relation independently of the validity correlation (Section 3.3.11 in [11]) but which may relate to differences in alcohol consumption (Section 3.3.13 in [11]).

*Duration of follow-up.* The number of dietary assessments was not independent of the duration of follow-up since the former were undertaken at regular intervals during follow up. In a multivariate model, the number of years of follow-up was shown previously not to be a significant factor affecting the size of the T2D-GL risk relation [17]. By contrast, it appeared previously that the duration of follow was a highly significant factor that could affect the size or the T2D-GI risk relation [20], an observation reproduced and Section 3.2.8 in [11]. However, more reliable evidence from within studies shows the duration of follow up not to be a statistically significant source of inconsistency (Section 3.2.8 in [11]).

*Number of dietary assessment.* This had no statistically significant effect of the size T2D-GI risk relation when analyzed within studies (Section 3.2.8 in [11]). Nevertheless RR tended to rise with increasing number of dietary assessments whereas our expectation was that should dietary assessment have taken place at baseline only, the T2D-GI risk relation would have been attenuated with time as the diet consumed diverged from that eaten at baseline [25].

Over all these sources of apparent inconsistency, the validity correlation for carbohydrate was dominant. Attention to the adequacy of dietary instruments therefore remains an important part of nutritional epidemiology. It seems from most cohort studies that dietary instruments are considered to be valid just because have been through a process of validation, rather than being put through a process of validation and reaching some pre-specified level of validity.

#### 3.2.4. Peoples, Places, Times—Circumstances

Statistically significant T2D-GI and GL risk relations have been observed in both women-only and men-only prospective cohort studies applying dietary instruments with validity correlations > 0.55 or adjusting for the validity correlation (centered on 0.7) (Tables 1, 4 and 5 in [11]).

By EQM, owing to the lack of quantitative information on GI and GL intakes preventing DRM, significant risk ratios were evident also for the women of different BMI (Sections 3.2.10 and 3.3.12 in [11]) although in the lower BMI category the lower 95% confidence limits did not meet the >1.10 criterion.

The T2D-GI risk relation arose in studies from different places, countries and continents including USA, China, Japan and Australia, with different ancestral ethnicities, from European and East Asia (Sections 3.2.16 and 3.3.18 in [11]). The observations, thus, were in different cohorts of people expected to have different genetic background and consume different foods.

In addition, in the USA, single-sex cohorts of women and cohorts of men (NHS 1, NHS II and HPFS) have been followed up for different periods of time. The published T2D-GI risk ratios were ≥1.20 [19,30,31,32,33], and presently estimated risk relations from dose-response meta-analysis were also >1.20 (Appendix A herewith). These observations suggest the risk relations occur over the different durations of time (and related ages of participants).

It is important to realize the Bradford Hill suggests making the same (similar) observations under “different” circumstances, not ‘all’ circumstances. Here for example, observations on the T2D-GI and GL risk relations met the criteria for public health interest in persons of European ancestry and East Asian ancestry. There was insufficient information on the strength of these relations in persons of South American ancestry and African ancestry, but as Hill would say, this is not proof of no T2D-GI or GL risk relation but would support the case for causation.


*Criterion conclusion:*
When robust approaches to data synthesis are used, the T2D-GI and GL risk relations (or risk ratios for BMI) among prospective cohort studies are sufficiently consistent both without and with adjustment for validity correlations to support a conclusion that the risk relations are of biological significance. The risks to health occur to a greater or lesser extent under different circumstances, e.g., different ethnic ancestry, places, times, foods, in addition to men, women, and higher (and possibly lower) BMI sub-populations of women.


### 3.3. Specificity

GI and GL are not the only possible determinants of T2D. It is important to establish, therefore, that other determinants were not the cause of the T2D-GI and GL risk relations observed in the original prospective cohort studies, i.e., that GI and GL are not surrogates for other factors. These factors can be divided into non-dietary and dietary factors.

#### 3.3.1. Non-Dietary Factors

Concern may arise that the T2D-GI and GL risk relations may be high due to being a surrogate for other factors known or thought to be a risk of T2D. However, study groups adjusting for such factors at the study-level have been addressed.

For the T2D-GI relation, the three largest prospective cohort studies (each >60,000 persons [19,30,34] and were from China and the USA (NHS I, NHS II) gave a random effects combined RR of 1.29 (1.11–1.50) (Table 1 in [11]]). This was similar to the RR for all 10 studies with CORR > 0.55 at 1.27 (1.15–1.40) (Table 7 in [11]) and for all 15 studies after adjustment for CORR (centered on 0.7) and FHD (centered on 0.5) at 1.28 (1.23–1.34) (Table 8 row 3 for GI in [11]). The three largest studies, adjusted for age, BMI, smoking status, physical activity and family history of diabetes.

From the 10 studies with CORR > 0.55, the T2D-GI risk relations ranged from 1.26 to 1.38 per 10 units GI for study sub-groups making study-level adjustments for age (*n* = 10 studies), smoking (*n* = 9), alcohol (*n* = 9), body mass index (*n* = 8), physical activity (*n* = 9), family history of diabetes (*n* = 8) and menopausal status in women and related hormone use plus oral contraceptive use (*n* = 2) (Table S17 in [11]). Grouping by study-level adjustment for level of education resulted in the lower T2D-GI risk relation of 1.17 (1.11–1.24) (*n* = 3) but this was attributable to variance among studies due to different values for CORR and FHD. Adjusting to common values for CORR (centered on 0.7) and FHD (centered on 0.5) all these groups gave a narrower range of T2D-GI risk relations, from 1.28 to 1.30 (Table S17 in [11]); this while allowing incorporation of studies with CORR < 0.55 (since CORR was a covariate). For most (75%) of these sub-groups, inconsistency (I^2^) within groups was zero with the remaining 2 groups (25%) having non-significant inconsistencies (Table S17 in [11]).

For the T2D-GL relation, the three largest prospective cohort studies (each >60,000 persons [30,33,34]) gave a random effects combined RR of 1.29 (1.19–1.39) (Table 4 row 8 in [11]), which was similar to the RR for all 15 studies with CORR > 0.55 at 1.26 (1.15–1.37) (Table 4 row 2 in [11]) and for 21 studies after adjustment for CORR (centered on 0.7) and FHD (centered on 0.5) at 1.34 (1.24–1.46) (Table 8 row 3 for GL and Table 4 row 5 in [11]). As noted above, these studies, adjusted for age, BMI, smoking status, physical activity and family history of diabetes.

From the 15 studies with CORR > 0.55, the T2D-GL risk relations ranged from 1.26 to 1.41 per 80 g/d GL in 2000 kcal diet when studies were grouped separately by having made study-level adjustments for age (*n* = 15 studies), smoking (*n* = 9), alcohol (*n* = 9), body mass index (*n* = 14), physical activity (*n* = 13), family history of diabetes (*n* = 7), menopausal status in women and related hormone use plus oral contraceptive use (*n* = 2) and level of education (*n* = 6) (Table S18 in [11]). Adjusting to common values for CORR (centered on 0.7 ) and FHD (centered on 0.5) gave all these groups a T2D-GL risk relations within the range 1.28 to 1.47, while allowing incorporation of studies with CORR < 0.55 (since CORR was a covariate) and, for most (88%) of these groups, reduced inconsistency (I^2^) within groups to ≤1% with the remaining 12% of these groups having non-significant inconsistencies (Table S18 in [11]).

These observations clearly show specificity of the T2D-risk relations to GI and GL rather than to these other known or considered risk factors for T2D. That is, GI and GL were not surrogates for these non-nutrient factors.

#### 3.3.2. Dietary Factors: GI and Dietary Factors in General

Among studies using a dietary instrument with CORR > 0.55, for which the combined T2D-GI risk relation was 1.27 (1.15–1.50) (Figure 3 in [11]) adjustments for dietary fiber of any type, cereal fiber, vegetable fiber, magnesium, protein, red meat, alcohol, energy, saturated fatty acids and *trans* fatty acids, the T2D-GI risk relations ranged from 1.27 to 1.50 per 10 units GI (Table 2 in [11]). Thus GI was not evident as a surrogate for any of these dietary factors.

Among dietary factors, particular concern has arisen prior to our meta-analysis for interpretation of the size of the T2D-GL relation with respect to fiber and protein. Therefore, we considered these factors further in Sections 3.3.3 and 3.3.4.

#### 3.3.3. Dietary Factors: GI and Fiber or Cereal Fiber

Comment is warranted in regard GI and dietary fiber(s) because of speculation that a risk relation due to GI may be due to fiber, particularly cereal fiber because neither fruit fiber nor vegetable fiber show an appreciable association with T2D risk [35].

Three large prospective cohort studies (NHS I [30], NHS II [19], and HPFS [19]) showed the T2D-GI risk ratios to be independent of the cereal fiber intake. Our dose-response meta-analysis of these data yielded a random effects T2D-GI risk relation of 1.39 (1.26–1.53) per 10 units GI (Table 2 in [11]). Notably, all three studies had used dietary instruments with validity correlations > 0.55. Previously it was shown [19] that lower cereal fiber and higher GI additively raised the risk of T2D, such that the joint risk ratio across tertiles was 1.59 [19], which corresponds to 1.74 (i.e., exp(ln(1.59•1.20))) across quintiles if log-linear.

From among eligible prospective cohort studies with CORR > 0.55 for which the combined mean random effects dose-response T2D-GI risk relation was 1.27 (1.15–1.40) (*n* = 10) (our primary outcome in [11]), seven studies adjusted for any type of dietary fiber (dietary, cereal and vegetable). The combined random effects T2D-GI risk relation from these eight studies was 1.31 (1.17–1.47) (Table 2 in [11]), again showing that the T2D-GI risk relation reported by studies was independent of fiber.

These results from prospective cohort studies are not surprising given that the glycemic index of foods correlates poorly (*r* = 0.17, *P* = 0.07) with dietary fiber content of foods [36]. Moreover, meta-analysis of intervention studies has also shown relations with GI and dietary fiber to be independent of one another, and additive, for the reversal of T2D risk factors: fasting blood glucose and glycated protein [37].

#### 3.3.4. Dietary Factors: GI and Protein

Foods with a low GI tend to be higher in protein content (*r* = −0.45, *P* < 0.001) [36]. Nevertheless T2D-GI risk ratios have been reported to be independent of protein intake [19]. Similarly, the T2D-GI risk relation of 1.39 (1.26–1.53) per 10 units GI for the 3 largest USA prospective cohorts studies (NHS I [30] NHS II [19] and HPFS [19]) occurred independently of protein intake (Table 2 in [11]).

#### 3.3.5. Dietary Factors: GL and Dietary Factors in General

Among studies using a dietary instrument with CORR > 0.55, for which the combined T2D-GL risk relation was 1.26 (1.15–1.37) (Figure 7 in [11]), adjustments for dietary fiber of any type, cereal fiber, magnesium, red meat, alcohol, energy, saturated fatty acids and *trans* fatty acids, resulted in sub-group T2D-GL risk relations that ranged from 1.26 to 1.56 per 80 g GL in 2000 kcal diet (Table S16 in [11]). Hence GL was not a surrogate for any of these factors. An exception was the adjustment for protein in 3 studies in which the T2D-GL risk relation was 2.02 (1.11–1.37) (Table S16 in [11]). However, these had a negligible impact on the overall T2D-GL risk relations because they represented just 3% of the weight of evidence. Among dietary factors, particular prior concern has arisen for interpretation of the size of the T2D-GL relation with respect to fiber, protein and alcohol. Therefore, we considered these factors further in Sections 3.3.6 and 3.3.7. 

#### 3.3.6. Dietary Factors: GL and Fiber

Uncertainty in evaluating the size of the T2D-GL risk relation sometimes arises from speculation that prospective cohort studies may not have adjusted for fiber intake [38]. However, when finding a T2D-GL risk relation was 1.26 (1.15−1.37) (*n* = 15, *P* < 0.001) (Figure 7 in [11]) several studies using a valid dietary instrument (CORR > 0.55) had either adjusted for dietary fiber [39,40,41,42] or cereal fiber [30,31,32,33,43] or shown during sensitivity analysis that adjustment for fiber made little difference [44,45]. Only one [46] of the 15 eligible studies with CORR > 0.55 did not report or comment on this aspect. Within the 15 eligible studies, those adjusting for dietary fiber or cereal fiber in their most adjusted model totaled 6 studies, for which the mean risk relation was no less than for all 15 studies combined and was 1.31 (1.00–1.75) (Table S16 in [11]). As for GI (Table 2) there was, therefore, no evidence that GL was a surrogate for fiber. However, this may not account for differences in the average population fibre intake among the studies.

When adjusting for CORR, 22 studies were available for meta-analysis of which 21 studies reported on fiber intake. Assessment of the sensitivity of the T2D-GL risk relation using dose-response meta-analysis, when adjusted for CORR (centered on 0.7, so allowing studies with CORR < 0.55 to be included), SEX of participants (centered on 0.5) and ETH (ethnicities: European-American vs. Others) (cf [17]) but without adjustment for population mean or median fiber intakes resulted in a T2D-GL risk relation of 1.34 (1.22–1.46) (*P* < 0.001, *n* = 22) (I^2^ = 0, τ^2^ = 0, *P* = 0.468) per 80 g/d GL in 2000 kcal diet). Additional adjustment for the sample population average fiber intake (centered on the mean 19 g/d sd 5, range 10 to 30 g/d and assuming 19 g/d in the study in which fiber intake was unknown) resulted in a T2D-GL risk relation also of 1.34 (1.22–1.46) (*P* < 0.001, *n* = 22) (I^2^ = 0, τ^2^ = 0, *P* = 0.505), which was identical to that without the adjustment for fiber (other than for the P-value for I^2^ = 0, τ^2^ = 0). The covariate for differences in fiber intake among studies had no significant effect (*P* = 0.214), though tended to show a slightly higher T2D-GL risk relations with lower fiber intake, again indicating GL was not a surrogate for fiber.

Moreover, for GL in addition to GI, it is important to note that meta-analyses of published studies has shown T2D not to be appreciably associated with the amount of either fruit or vegetable fiber consumed [35,47]. This fiber could therefore make little impact on the T2D-GL risk relation. By contrast with fruit and vegetable fiber, incident T2D is clearly associated with cereal fiber intake [35], however, the T2D-cereal fiber and T2D-GL relative risks have been shown to be independent of one another and additive [19,31,32,47].

Further, the independence and additivity of effects of GL and dietary fiber found in prospective cohort studies, have been demonstrated by meta-analysis also for both fasting blood glucose and glycated proteins in intervention studies. Thus both lower GL and high dietary fiber can individually and additively reverse these risk factors for T2D [37]. Taking all this evidence together, there was none that showed fiber consumption significantly confounds the T2D-GL risk relation.

#### 3.3.7. Dietary Factors: GL, Alcohol and Protein

Protein as a putative covariate marker of conditional carbohydrate intake (endnote b in ref [11]) occurred in only 3% of the weight of evidence (*n* = 22 studies with adjustment for CORR, SEX, ETH (American-European compared to other ethnicities) and FUY (number of follow-up years) as in [17]) and so had negligible impact on the size of our combined studies overall T2D-GI risk relation (Table 5 row 4 in [11]).

Higher population average alcohol consumption among prospective cohort studies was significantly (*P* = 0.039) associated with a lower T2D-GI risk relation (Figure 9 in [11]) when simultaneously adjusted for CORR, ETH and FUY. However, this had negligible or no effect on the overall combined mean T2D-GL risk relation, which was 1.31 (1.19–1.44) (Table 5 row 5 in [1]) rather than 1.33 (1.21–1.45) per 80 g/d GL in 2000 kcal (8400 kJ) diet (Table 5 row 1 in [1]). That the covariate for SEX could be replaced by the covariate for alcohol raised the possibility that attenuation by alcohol may simply be due to a sex-associated difference in alcohol consumption. Alternatively, alcohol is known to acutely reduce gluconeogenesis and moderately reduce hepatic glucose production in the fasting state [48] and when consumed before a meal alcohol reduces the glycemic response to a meal [49] so may in part mimic a lower glycemic load.


*Criterion conclusions:*
Considering all eligible prospective cohort studies on GI or GL together and recognizing the potential for residual confounding, major non-dietary factors were unable to explain the strength of association between T2D and GI or GL. The non-dietary factors included age, race, weight, smoking status, physical activity and family history of diabetes, as well as menopausal status and use of post-menopausal hormonal therapy in studies of women.Similarly, among dietary factors, intakes of total energy, *trans*-FAs, SFAs, protein, fiber or cereal fiber and alcohol in the original prospective cohort studies do not explain the strength of study-level associations between T2D and GI or GL.T2D-GI and GL risk relations and T2D-fiber (or cereal fiber) risk relations are independent and additive.Alcohol intake appears to attenuate the T2D-GL risk relation, thus a sex-difference in alcohol consumption may explain a sex-difference in the strength of their T2D-GL relations.The strength of the T2D-GL risk relation found is independent of potential confounding by simultaneous adjustments for intakes of energy and multiple macronutrients including protein.


### 3.4. Temporality

Both prospective cohort studies [11] and intervention studies (considered below) exist for GI and GL and each study type contributes to establishment of temporal relations. At the present time, no intervention study has been published that examines whether consuming reduced GI or GL foods can reduce incident T2D as an *endpoint*. However, intervention studies in both Caucasian and Asian populations have been conducted using alpha-glucosidase inhibitors of intestinal starch and sucrose digestion (e.g., Acarbose) to slow (rather than prevent) carbohydrate digestion and lower the glycemic response to diet, thereby lowering dietary GI and GL [50,51,52]. This was accompanied by a dose-dependent reduction in the incidence of T2D among persons with impaired glucose tolerance (see *Analogy*, Section 3.8). Further, meta-analyses of several intervention studies that replace higher with lower GI carbohydrate foods have shown temporal relations independently of dietary fiber for key surrogate end-points among persons with diabetes—notably for fasting blood glucose and glycated proteins, consistent with reversing the progression of T2D [37,53] (see *Experimental evidence*, Section 3.7). In the prospective cohort studies, incident T2D was higher post baseline with higher dietary GI or GL during 4 to 26 y-follow-up in [11]) (see *Strength of association*, Section 3.1). In addition, a prospective cohort study has linked the glycemic components of metabolic syndrome with dietary GI and GL [54]. In regard to the prospective cohort studies, it is unlikely that persons progressing to metabolic syndrome and T2D could consciously decrease their dietary GI or GL values because most people are unaware of what constitutes a diet low in GI or GL—at least in most parts of the world at the time the studies were conducted.


*Criterion conclusion:*


A temporal relationship of GI and GL to prevent or delay T2D is indicated by 3 independent sources of data:Prospective cohort studies in which incident T2D occurs after consumption of the diets different in GI or GL.Randomized controlled intervention trials that show plausible mechanisms and relevant changes in T2D risk factors.Randomized controlled intervention trials that use tolerable doses of alpha-glucosidase inhibitors (e.g., Acarbose) to slow rather than prevent carbohydrate digestion in the small intestine (thereby lowering dietary GI or GL) result in lower or delayed incidence of T2D. These inhibitors act only in the gut and are not absorbed into the circulation.

### 3.5. Biological Gradient (dose-dependency)

Prospective cohort studies provide evidence of non-linear dose-response risk relations for T2D with GI and GL when assessed over the global (worldwide) range of GI and GL intakes [11]. The random effects T2D-GI risk relation across global intakes of GI (47 to 76 units GI) reached 1.87 (1.56–2.25) (*P* < 0.001, *n* = 10) and the random T2D-GL risk relation across global intakes of GL (73 to 257 g/d in 2000 kcal diet) reached to 1.89 (1.66–2.16) (*P* < 0.001, *n* = 15). Both relations (per 10 units GI and 80 g/d GL in 2000 kcal diet) were stronger at lower intakes than at higher intakes of GI and GL [11]. However, usually it is the combined dose-responses within population jurisdictions, satisfied by local restricted cubic spline (non-linear) dose-response meta-analysis that have been of interest for guidance on Public Health.

For the local dose-response meta-analysis of the T2D-GI risk relation using prospective cohort studies, only those studies applying dietary instruments with validity correlations for carbohydrate > 0.55 and reported results by 3 or more quantiles (or doses) were used, as cited in Figure 1 legend. This also provided clear evidence of a biological gradient and no evidence of departure from log-linearity (*P* = 0.989 for difference for the red curve vs. the black curve in Figure 1). The random effects T2D-GI risk relation increased by 1.32 (1.25–1.40) (*n* = 8, *P* < 0.001) per 10 units GI. whereas the random effects T2D-risk relation using all 10 studies with a validity correlation > 0.55 was 1.27 (1.15–1.40) by two-step DRM (Figure 3 in [11]).

For the local dose-response meta-analysis of the T2D-GL risk relation using prospective cohort studies, again only those studies applying dietary instruments with validity correlations for carbohydrate > 0.55 and reported results by 3 or more quantiles were used. These comprised 13 studies (from 10 publication, Figure 2 legend). The 13 studies are those in Figure 7 in [11] with validity correlations > 0.55 but excluding the study of Patel et al. [46] and Sluijs et al. [40] because the paper reported on <3 quantiles for GL. From the 13 studies there was clear evidence of a biological gradient and no evidence of statistically significant departure from log-linearity for GI (red curve vs. black curve in RR, P-value for difference = 0.194, Figure 2). The random effects T2D-GL risk relation assuming linearity (red curve in Fig 2) increased by 1.27 (1.22–1.36) (*n* = 14, *P* < 0.001), whereas the random effects T2D-risk relation using all 15 studies with a validity correlation > 0.55 was 1.26 (1.15–1.37) by two-step DRM (Figure 7 in [11]).


*Criterion conclusion:*
Highly powered prospective cohort studies and dose-response meta-analyses show that both the T2D-GI and the T2D-GL relations are dose-dependent over a wide range of GI and GL values, both locally and globally.


### 3.6. Plausibility (Mechanisms)

Evidence of plausibility is helpful and important to the establishment of confidence in causation [5]. The central mechanisms underlying causation (and potentially management) of T2D in relation to GI have been summarized [2,3]. T2D develops in individuals who experience both insulin resistance and beta-cell dysfunction which eventually leads to reduced beta-cell mass. Several pathways lead to *endpoints* of reduced risk of insulin resistance and β-cell dysfunction, i.e., the pathologies underlying T2D. One pathway is: Reduced GI of foods eaten → decreased hyperglycemia → deceased hyperinsulinemia and beta-cell demand → decreased free-fatty acids → decreased *endpoints*, which has can been called the lipotoxic route [3]. A second and shorter sequence is: Reduced GI of foods eaten → decreased hyperglycemia → decreased *endpoints*, in which glucose has adverse effects and which has been called the glucotoxic route [3]. Finally, a third route has been suggested: [2] Reduced GI of foods eaten → decreased body weight → decreased *endpoints*, which might be called the ponderal route. The same pathways are implicated for GL [2,3].

#### 3.6.1. Glucotoxicity and Lipotoxicity, Including Inflammation

Both the glucotoxic and lipotoxic routes elicit several putative intermediary mechanisms. The routes from higher glycemia and higher free-fatty acids to the *endpoints* were established well over a decade ago [2,3,56] or even longer [57] and now provide targets for treatment strategies [4]. Included in these ‘toxic’ processes is a role for post-prandial hyperglycemia in the generation of oxidative-free radicals, cell death, and the release of pro-inflammatory cytokines and adhesion molecules that contribute to beta-cell dysfunction and apoptosis [4,58]. The inflammatory response triggers the development of insulin resistance, metabolic syndrome and T2D [59]. Each process is evident in both obese (non-diabetic) persons and persons with diabetic complications (endothelial dysfunction and cardiovascular disease) [58,60].

Instrumental to completing these mechanistic links, from GI to the *endpoints*, are: (a) That both normal and hyperglycemic persons (e.g., T2D) respond with even higher glycemia postprandially after eating higher rather than lower GI foods [61,62,63]. (b) That from meta-analyses of controlled intervention studies, higher GI foods are less effective than lower GI foods in studies up to 3 months duration for lowering of fasting hyperglycemia (and glycated proteins, HbA_1c_ and fructosamine) [37,53,64], with the largest absolute difference seen in persons having the most severe fasting hyperglycemia [37]. (c) That in the late post-prandial phase, higher GI foods elevate free-fatty acids due to a counter regulatory hormone response to excessive insulin [65,66,67,68,69,70]. Further to this, lower free-fatty acid concentrations have been observed throughout the day when consuming diets lower in GI [66], and over several hours when slow absorption due to sipping glucose rather than taking the same amount as an oral bolus [57]. (d) GI of carbohydrate foods is inversely associated with liver fat in insulin-resistant subjects (see Section 3.6.2
*Ponderal toxicity* below), possibly through mechanisms related to the reciprocal control of glucose and fatty acid metabolism in the liver and other tissues [71]. Liver fat accumulation leads to non-alcoholic fatty liver disease (NAFLD), which beside its direct effect on health, represents an independent risk factor for T2D [72]. Lastly, (e) Meta-analysis indicates inflammatory markers including C-reactive protein are predictive of T2D [73] though possibly not independently of baseline blood glucose or fatty liver [74] with both glucotoxicity and lipotoxicity being involved [4].

A systematic review has revealed that epidemiological and a majority of randomized, controlled studies show lower glycemic diets (whether in GI, in GL, or GI and GL) associate with lower levels of C-reactive protein, with only a minority of studies showing no significant effect [59]. Taking all relevant studies together, the effect on C-reactive protein was found significant (*P* = 0.01) by meta-analysis [75]. This finding for GI has since been confirmed in two large controlled studies of 6 months duration in persons without diabetes and of 12 months duration in T2D patients [76,77].

#### 3.6.2. Ponderal Toxicity

Obesity, particularly central obesity poses a risk of T2D [78,79]. Reduction of central obesity is accompanied by normalization of insulin secretion and hepatic insulin sensitivity [80], thereby reversing the underlying biochemistry of pre-diabetes. Diets of higher GI or GL foods may stress body weight (or body composition) regulation in addition to the gluco- and lipo-regulatory mechanisms. Those involved in conversion of excess body weight (or adiposity) to T2D have been discussed previously [3,81] and again include gluco- and lipotoxicity. Included is lowering the glycemic response, hunger and central obesity further by incretin responses to low-GI carbohydrate foods [82,83]. Further, an overlapping route proposed involves high GI (rapidly absorbed) carbohydrate elevating glucose-induced insulinotropic peptide (GIP) which promotes lipogenesis, fatty liver, insulin resistance and postprandial inflammation [84].

Meta-analysis has shown a sufficient reduction in dietary GI or GL does result in a dose-dependent reduction in body weight in free-living persons with diabetes without them attempting to control their food intake [37]. Reduced glycemic load by avoidance of high GI carbohydrate foods has also resulted in the loss of body weight and gaining control of T2D [85,86,87]. Lower GI diets have also helped body weight reduction and maintenance, particularly with a background diet high in protein [88,89,90].

Meta-analyses of long-term studies has also indicated low GI or GL diets can reduce body fat mass in obese patients [75], in the shorter term lower abdominal (central/visceral) adiposity in adults [63,91,92,93,94,95], lower liver fat [71] and lower a serum marker of non-alcoholic fatty liver disease [87]. Reduction of central obesity, particularly hepatic fat, is of major importance [78,79].

Further evidence comes from studies of women during pregnancy. Low GI diets were found to limit the incidence of excessive birth weight (macrosomia, unrelated to fat mass), and importantly to reduce the use of insulin in those with gestational diabetes mellitus [96].

In summary, consuming diets lower in GI, and possibly more so diets lower in GL, can be accompanied by lower body weight and reduced central obesity. Where considered preferable, GI provides a means to lower the glycemic response to diet without having to lower carbohydrate intake. However, a reduced intake of higher GI carbohydrates in the context of lower carbohydrate lower energy diets may be preferable in overweight and obese persons with prediabetes or T2D [85,86,87]. While helpful, body weight or fat reduction without deliberate or intentional reduction of energy intake is modest, and long-term sustainability remains uncertain.


*Criterion conclusions:*
At least three complementary mechanistic chains of events link higher GI and GL to T2D in a causative manner: these include elevation of glucotoxicity, lipotoxicity, and ponderal toxicity including central obesity. All three lead to compromised beta-cell function.Effects on weight. Weight outcomes are modest if related solely to effects of GI or GL on the rate of weight loss. Effects on central obesity may arise in the absence of significant body weight change. Limiting the intake of high GI foods in the context of limiting carbohydrate intake may support both body weight and central obesity reduction.


### 3.7. Experimental Evidence

Even though controversy exists over the interpretation of rodent studies, early studies in laboratory rats have provided complementary evidence for higher rather than lower GI carbohydrate causing increased adiposity, decreased lean body mass, glucose intolerance, and disruption of islet-cell architecture independently of body weight and dietary macro- and micro-nutrient composition [97]. As summarized by Chaumontet and colleagues [98] when reviewing studies in rats [99,100,101,102,103,104]—higher GI carbohydrates cause: (a) impaired lipid oxidation that occurs in advance of insulin resistance and higher insulin secretion, (b) development of insulin resistance in advance of glucose intolerance, (c) in the longer-term development of glucose intolerance and hyperinsulinemia, and (d) development of obesity, visceral adiposity and elevated liver triglyceride.

Similar observations were made in controlled intervention studies in humans, in particular higher GI carbohydrates causing: (a) lower glucose tolerance as a second meal effect without elevation of colonic fermentation [105,106,107]. The same outcome was demonstrated also for glucose taken by mouth as a bolus compared with continuously sipped glucose to mimic the slow absorption of glucose from low GI foods [57], (b) lower insulin sensitivity in a meta-analysis of 18 intervention studies [combined mean −20% (95% CI 6, 33; *P* = 0.004)] with similar results at <12 and >12 weeks treatment duration [37], (c) poorer islet β-cell function [108,109], and d) elevation of abdominal or visceral adiposity [63,91,92,93,94]. Limiting the intake of high-GI carbohydrate (i.e., moderately decreasing GL) has also improved liver function as measured by serum ϒ-glutamyl-transferase in recently diagnosed glucose intolerant and T2D patients, suggesting a reduction in fatty liver or non-alcoholic fatty liver disease [86].

There is still uncertainty of the benefit of a low GI diet among persons without diabetes due to inconsistent results on fasting blood glucose [110]. A published meta-analysis [37] indicated that lower GI diets normalize fasting blood glucose concentration in persons with fasting glucose >6 mmol/L. However, in those with baseline glucose <~5.5 mmol/L, lower GI diets appeared to modestly elevate the fasting blood glucose relative to controls (see Figures 1 and 2 in [37]). This paradox is difficult to explain fully because the effects on fasting insulin were relatively small (Figure 7 in [37]). However, according to the homeostatic model assessment of insulin resistance [HOMA-IR] [111,112], sensitivity to insulin would improve among persons consuming lower GI vs. higher GI diets across a range of fasting blood glucose values >6 mmol/L. The outcome of this scenario has been confirmed in more recent studies with treatment durations from 3 to 18 months [110,113,114,115,116,117,118]. Meta-regression of these study results showed a transition point for the direction of effect on fasting glucose at 5.4 mmol/L (Appendix A herewith), which is consistent with regular consumption of lower GI rather than higher GI carbohydrate foods guarding against moderate hypoglycaemia. Data such as these support improvement in glycemic control irrespective of status of blood glucose control, from normal to the T2D condition.


*Criterion conclusion:*
Experimental studies in animals and humans show diets of higher GI and GL cause significant features of T2D while diets of lower GI or GL show the reverse.


### 3.8. Analogy

GI, GL and the glycemic response to foods are lowered by oral intake of inhibitors specific to carbohydrate digestion in humans—acarbose and voglibose (alpha-glucosidase inhibitors) [50,51,52]. Primarily, these inhibitors slow carbohydrate digestion in the small intestine rather than blocking it, although a small amount of carbohydrate escapes digestion and is fermented in the large bowel. The inhibitors themselves are not absorbed into the bloodstream. Two large, long-term (>3 y) randomized controlled trials in persons with pre-diabetes have been conducted, one in Japan with voglibose [52] and one in Canada with acarbose [50,51]. Both studies showed prevention or significant delay of incident T2D thereby reducing the prevalence of this disease. This is in agreement with the underlying principle that carbohydrate foods causing a low glycemic response, as characterized by low GI or GL, are beneficial among persons with pre-diabetes. The study with acarbose also showed significant reversion of impaired glucose tolerance to normal glucose tolerance (*P* < 0.0001). While the study protocol reduced the likelihood of gastrointestinal symptoms, higher rates of large bowel fermentation were evident [119,120,121,122]. However, increased fermentation and changes in the microbiome may also contribute to beneficial effects of these drugs, and this can apply to low GI diets [123]. Notably, the reduced incidence of T2D due to use of acarbose was independent of changes in body weight, and like low GI and low GL diets, improved a range of biological markers of risk of T2D [124].


*Criterion conclusion:*
Evidence from randomized controlled trials indicate that inhibitors used to slow carbohydrate digestion (analogous to lowering dietary GI and GL) prevents or delays progression of impaired glucose tolerance to diabetes.


### 3.9. Coherence

Coherence with the natural history and biology of disease is evident from Section 3.6 and Section 3.7 on *Plausibility* and *Experimental*. Here we examined coherence from additional standpoints, in relation to surrogate markers for T2D and the epidemiology of diseases associated with T2D.

#### 3.9.1. Surrogate Markers for T2D

Ingestion of the alpha-glucosidase inhibitor acarbose has been shown to improve a range of markers for T2D and CHD risk. These changes are similar in mechanism and magnitude to those seen with diets of lower GI/GL. Both interventions lower the postprandial glycemic response and favorably affect the risk markers. Acarbose also prevents or delays the development of T2D among persons with impaired glucose tolerance [124]. In addition, the T2D-GI and T2D-GL risk relations in prospective cohort studies have independent and additive association with the T2D-dietary fiber risk relation (see *Specificity*, Section 3.3). Likewise interventions also show independent and additive effects of GI (and GL) and dietary fiber on two important surrogate markers of the severity of T2D, namely fasting glucose and glycated proteins [37]. Notably, increasing the GI without change in fiber content (e.g., by processing and reduction in particle size) also led to worsening of markers for glycemic control and lipid profiles [125].

#### 3.9.2. Body Mass Index (BMI)

T2D can occur among persons of any BMI (kg/m^2^) while risk of T2D increases with BMI [126,127,128]. Hence we might expect higher GI and GL to increase the risk of T2D at all levels of BMI. In our meta-analyses, association of incident T2D with higher GI and GL was evident in lower and higher BMI strata. In both cases RR was ≥1.20 (see *Strength of association*, Section 3.1 herein and Sections 3.2.10 and 3.3.12 in [11]). When the T2D-GI (and GL) relation and the T2D-BMI relations were joined, remarkably high joint RR values of 12 (11–14) and 11 (10–12) have been reported for women in the USA [19] for GI and GL respectively. A joint relation for men in Japan of 7.1 (without confidence intervals) has also been published [41]. While these values may seem too high, they signal important effects of GI and GL together with body weight.

#### 3.9.3. Heart Disease

It is well established that patients with T2D are at a higher risks for coronary heart disease (CHD) and other cardiovascular disease (CVD), and that among persons with higher glycated hemoglobin, even without a diagnosis of diabetes, there is an increased risk of heart disease [129,130]. Should higher GI and GL promote T2D, then we might expect promotion of heart disease too. In agreement, published meta-analyses of results from prospective cohort studies have shown both GI and GL associate positively with both CHD risk (*n* = 10 studies) [14] and CVD risk (*n* = 14 studies) [131]. Bradford-Hill criteria indicate these relationships are probably causal [14,15,131].

In summary, data on the T2D-GI and GL relations herein are coherent with those published for the CHD/CVD-GI and GL risk relations.

#### 3.9.4. Cancers

Insulin resistance and hyperinsulinemia promote cancer cell growth [82,132,133]. Consistently, diabetes increases the risk of many common cancers, including liver, pancreas, bladder, breast and colon cancer [83,134]. If higher GI and GL diets promote T2D, then we might expect a higher risk of cancer generally. Indeed, meta-analyses of prospective cohort studies confirm this expectation. Notably, patients with T2D are at higher risk of colorectal cancer as evidenced by meta-analyses of 8 prospective cohort studies [135,136]. Meta-analysis of 15 prospective cohort and case-control studies also supports an association between both GI and GL and colorectal cancer, with moderately elevated RR values of 1.16 (1.07, 1.25) and 1.10 (0.97–1.25), respectively. Recent evidence on achieving lower GI or GL diets using an inhibitor of carbohydrate digestion, acarbose, is also consistent with higher GI or GL promoting colorectal cancer [137].

In summary, data on the T2D-GI and GL relations herein are coherent with those for colorectal cancer-GI and GL risk relations, both indicating greater risks consequent on a greater glycemic, and related insulinemic, responses.


*Criterion conclusions:*
*Surrogate markers of T2D:* Interventional evidence on surrogate markers of T2D risk and reversal of disease progression, as monitored by fasting blood glucose and glycated proteins, are coherent with robust observations on incident T2D-GI and GL risk relations.*BMI, CHD and cancer:* Both overweight and obese persons are at risk for T2D, and both are at lower risk when consuming diets lower in GI or GL. T2D associates with CHD and colorectal cancer, and persons consuming diets of lower GI and GL are also at lower risk for both of these conditions.


## 4. Discussion

In this analysis, all nine of the Bradford-Hill criteria were met (= 1 each) for GI and GL indicating that we can be confident of a role for both as causal factors contributing to incident T2D (Table 2). Hard proof of causation, i.e., very long-term intervention studies that subject individuals to the risk of consuming higher GI carbohydrate foods are unlikely to be conducted for ethical and financial reasons. By contrast, our meta-analyses of prospective cohort studies have provided observations from a natural setting with realistic changes in GI and GL that are more generalizable. Interventions can suffer from varied adherence to treatment and control over long durations especially when food preferences contrast with experimentally targeted food choices. In nutrition intervention studies, confounding goes unnoticed [37].

The criteria used (RR ≥ 1.20 or ≤ 0.83 in the right direction; Table 1) divides the strength of association into two categories in respect of public health; ‘sufficient strength’ and ‘insufficient strength’. The former identifies nutritional risk factors generally applicable in public health when other Bradford-Hill criteria are met, while the latter identifies nutritional risk factors that may not see application in nutrition guidelines. However, policy decisions may upgrade or downgrade acceptance for this application and may be subjective. These values (RR ≥ 1.20 or ≤ 0.83) were also used independently of us by Mente et al. [15] and dependently by Livesey and Livesey [14] when applying Bradford-Hill criteria to coronary heart disease-GI and GL risk relations. Choice of the these criteria is supported by recognition that a 20% increase or decrease in risk related to a nutrient or food or dietary pattern can be both biologically plausible and have a meaningful effect on public health as a result of a widespread exposure [138], which applies to dietary carbohydrate, GI and GL. As noted by others [138] a commonly cited criterion of relative risks < 2.0 being suspect does not necessarily apply to nutritional epidemiology. Moreover a nutritional relative risk > 1.50 is considered to be strong [138] thus the global (applying to the worldwide range of exposures) relative risk of approximately 1.9 for the T2D-GI and GL risk relations [11,20] and even higher global RR values for the CHD-GI and GL risk relations [14] can be considered to be strong in respect of causality assessment.

The corresponding least deviant 95% confidence limits for the criteria in respect of public health (>1.10 and <0.91 in Table 1) are also of similar strength and were adopted by us and subsequently used in [14] specifically to justify that nutrients in question make a clinically significant difference in risk; this reflects that a 10% difference in risk was consider clinically significant by a prior expert group [16].

A lesson from Bradford Hill himself [5] is that guidelines for public health can be made in the absence of hard proof whenever other evidence is sufficiently strong (as in the present instance), when there is likely to be no harm (as is the case here), when health cost saving are anticipated (as expected for GI and GL), and provided there is evidence from among the general public i.e., relevant epidemiology—as in the present instance. In the past, researchers have used Bradford-Hill criteria in multiple ways. Examples include: (a) communicating limited information in a few short paragraphs [139], which places a severe limitation on the presentation and discussion of the wider evidence; (b) choosing to report only those criteria thought most applicable [15,140], which in public health and medical nutrition therapy limits consideration and understanding of any need for applications in advice on foods, nutrients, and nutrition guidelines; and (c) by incorporation of significant parts of Bradford-Hill criteria into a grading system, GRADE [6], which again circumvents consideration of the wider but relevant evidence. The strength of the approach used here is that we retained all the criteria and reviewed each of them comprehensively. Finally, we have provided a view of the potential cost benefit [10], which falls outside the consideration of causality, but is important to the consideration of the utility of GI and GL in T2D prevention in public health.

The limitations of the Bradford Hill’s approach should also be noted. Causality is ultimately a judgment call. The criteria employed here are a useful and systematic approach for evaluation of the evidence [6,7,8,9,141] without need of proof, or assertion of proof via scaling of each criterion by subjective assessment of the strength of evidence. For strength of association, as an example, we have used a quantitative criterion that is both relative to the strength of association for other dietary exposure variables and pragmatic in that it matches the strength of evidence that public health authorities have adopted post hoc when drawing upon evidence from population based studies. In addition, particular attention has been given to potential confounders.

Taken together, our findings and analysis indicate that it is appropriate for authorities to offer food and nutrition advice that incorporates GI and GL considerations. Low GI and GL advice does not conflict with current healthy eating dietary advice and should be applied in the context of healthy diet, healthy food and nutrient-based advice. At present, conventional dietary advice is inadequate for identifying low GI or GL foods or diets.

It is notable, too, that low GI and GL food carbohydrate diets are compliant with healthy diets from sustainable healthy food systems [142]. That is food production is a major driver of land and water usage, CO_2_ emissions, reduced biodiversity and other environmental risks, while plant based and carbohydrate foods are proposed to be the cornerstone of healthy diets with lower than average impact on ecosystems and stability of Earth systems. Knowledge of the quality and amount of the carbohydrate foods we eat is important now and will be of increasingly importance in future years to achieve reductions in both non-communicable diseases and promote sustainable development.

The advice offered by various national and international authorities falls into two basic and complementary approaches. One is food-based and provides consumption guidance on the amounts of fruits, vegetables, wholegrains, etc. used with or without advice on dietary patterns. The second approach is nutrient-based and provides consumption guidance on the amounts of energy, protein, fat, carbohydrates, dietary fiber, and specified micronutrients. In effect, the advice gives an alternative hypothesis, that diets low in GI and GL would associate with incident T2D not because GI and GL are causative of T2D but because healthy diets as defined by the advice are both preventative of T2D and low in GI and GL. However, this alternative ignores the wide variation in GI of foods within food groups.

Food-based advice remains intact (unchallenged by results presented herein) but knowledge of the food GI or GL values would improve specific food choices within food categories communicated by food-based advice. This is important because food-based advice alone does not guarantee lower GI or GL choices from either food categories or diet composition. Thus, each food category describes foods with wide ranges of GI and GL values [143,144]. The ranges are so wide in each food-category that it is possible to adhere to food-based advice alone (including patterns) and still be able to choose foods of higher, medium or lower GI values from within each food category.

It is not surprising therefore that choosing fruits of lower compared to higher GI has led to lower risks factors for T2D and CHD [145]. Likewise not surprising is that among persons with a high score for adherence to a Mediterranean diet, a lower compared to higher GL Mediterranean diet predicts a lower incidence of T2D [146]. Further to this, in a prospective cohort study of consumers of vegetarian and healthy diets in the UK, those consuming a higher vs. lower GI diet, had a higher risk of T2D [Risk ratio = 1.33 (0.88, 2.02), without quantitative information on intakes] [147]. The Mediterranean and vegetarian diets are currently amongst the healthiest dietary patterns advised in public health. It is noteworthy, therefore, that lower GI or GL versions of these heathy diets have shown added benefit.

Advice about dietary fiber consumption to reduce the risk of incident T2D is not sacrificed by advice about GI or GL. This is because, both epidemiological evidence in persons without diabetes and interventional evidence in T2D patients show that lower GI and higher dietary or cereal fiber provide additive risk reduction for incident T2D and glycemic control, respectively [31,32,33,37]. Further, cereal fiber (and possibly total dietary fiber) might sometimes be effective only when diets are undesirably high in GI or GL. Thus, no T2D-cereal fiber relation was found in women consuming diets low in GI [33] and in men consuming diets low in GL [31]. These results in prospective cohort studies occurred even though cereal fiber provides greater protection than either fruit or vegetable fiber [32,33,148,149]. Consistently, an intervention to lower the GI of foods eaten in T2D patients was found preferable to an increased consumption of cereal fiber [150]. Note meanwhile, authors in [22] advocate (WHO) to continue with advice of dietary fiber in nutrition guidance while the combined relative risk they found for dietary fiber was 0.84 (0.78–0.90) (without reporting the range of dietary fiber intake to which this applied) (Table 1 in [22]). This result just fails to meet the first criterion (Table 1 herein) of a combined mean RR of <0.83 for beneficial nutrient when the referent was at the lowest category of exposure. Thus either further studies or better meta-analyses (see Section 3.1. herein) are still needed in regard T2D and dietary fiber.

Wholegrain consumption as a hypothetical measure of cereal fiber consumption is also not a predictor of a food’s GI [143,144]. Further, selection of wholegrain can be associated with a slightly higher GL and energy consumption instead of a more desirable lower GL and energy [151]. It is clear from these studies that neither dietary fiber nor cereal fiber nor wholegrain are reliable or effective surrogate measures or beneficial markers of GI or GL, as has sometimes been assumed.

Further, it is important, too, that habitual diets composed of lower GI or GL foods facilitate adherence to recommendations for most micronutrients [140,152]. In both clinical trials and observational studies, children, adolescents, and pregnant women consuming a diet lower in GI generally had a more nutritionally adequate diet [153,154]. However, as noted above, health advice on foods without attention to the GI and GL does not predict low GI or GL diets [143,144].

A systematic review shows that calculating the cost of diabetes is complex and inexact [155]. Internationally, treatment costs for diabetes may reach 10% of national health expenditures and the losses of national income can be similarly costly [156,157]. Potentially, approximately 20% of these costs may relate to the consumption of diets too high in GI or GL when considering an RR of ~1.26 (Sections 3.2.2 and 3.3.2 in [11]) where approximately 20% derives from 100–100/1.26. Meanwhile, existing treatment costs are escalating along with the increasing prevalence of diabetes [128] and there is concern that drugs alone may not prevent progression of diabetes complications [158].

## 5. Conclusions

Glycemic index and glycemic load are dietary factors probably causal of type 2 diabetes and should be considered by future dietary guideline committees for inclusion in food and nutrient-based recommendations.

## Figures and Tables

**Figure 1 nutrients-11-01436-f001:**
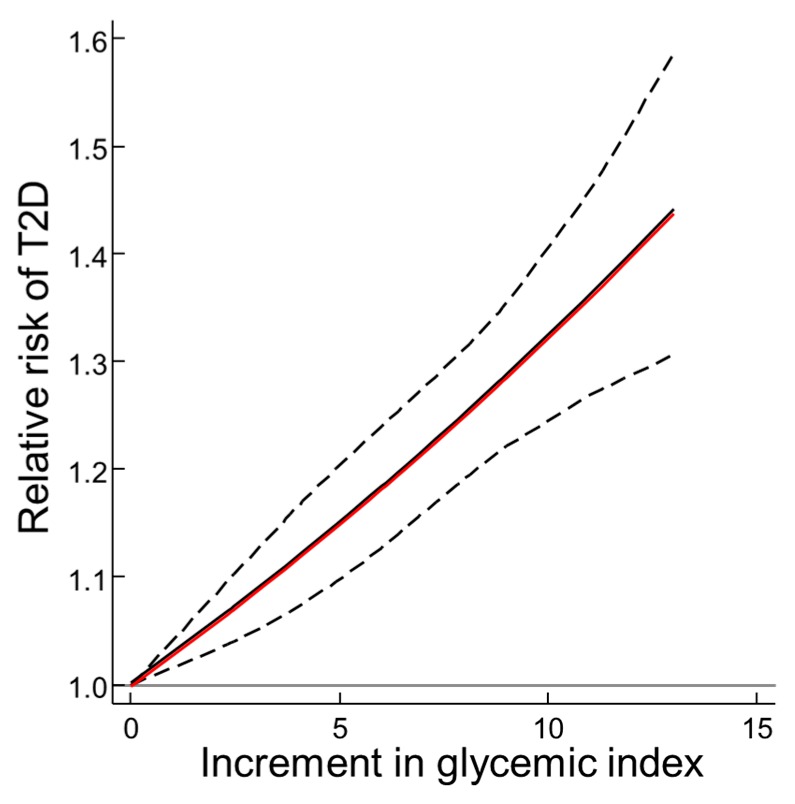
Local dose-dependence of the relative risk of T2D on Glycemic Index in prospective cohort studies combined. Analysis was on the log relative risk (RR) and is shown exponentiated to RR. All studies had a validity correlation coefficient for their dietary instrument > 0.55 for carbohydrate and reported by 3 or more quantiles. The slope (black continuous line) and 95% confidence intervals (dashed lines) derived from restricted (natural) cubic spline meta-regression (*glst*) with three knots at 0, 5 and 11 unit increment in GI. For comparison a log-linear dose response is also shown (red). There was no significant evidence for departure from log linearity (*P* > 0.989 for the secondary spline determinant). Thus the plots (red compared with black) overlapped consistently throughout the range. The log-linear dose-response T2D-GI risk relation rose by 32% per 10 unit increment in GI, i.e., RR = 1.32 (1.25–1.40), *P* < 0.001, *n* = 8 studies including 29 increments in GI. The mean GI at the intercept was 55 units GI based on the glucose (GI = 100) scale. The studies were Bhupathiraju et al. (2 studies, HPFS and NHS II) [19], Oba et al. in men [55], Mekary et al. (NHS I) [30], Sakurai et al. [41], Villegas et al. [34], Sahyoun et al. [45] and van Woudenbergh et al. [42]. Abbreviations: GI, glycemic index; RR, relative risk; T2D, type 2 diabetes.

**Figure 2 nutrients-11-01436-f002:**
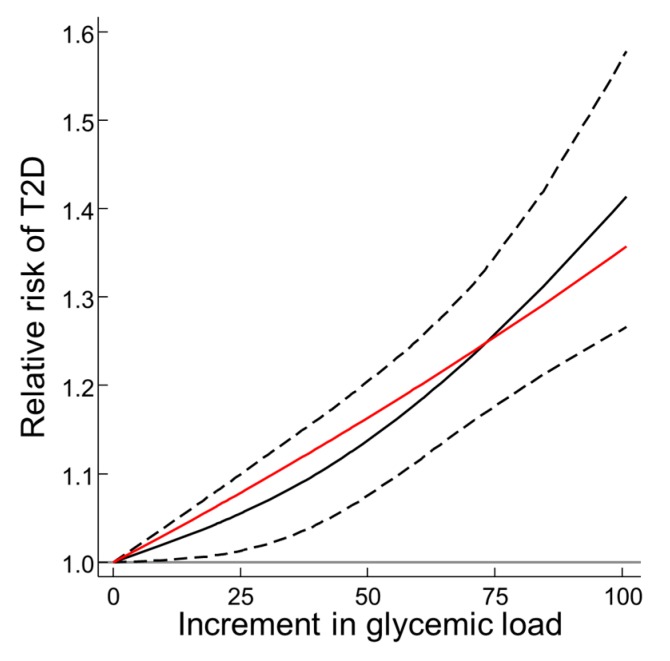
Local dose-dependence of the relative risk of T2D on Glycemic Load in prospective cohort studies combined. Analysis was on the log RR and is shown exponentiated to RR. All studies had a validity correlation coefficient for their dietary instrument of > 0.55 for carbohydrate and reported by 3 or more quantiles. The slope (black continuous line) and 95% confidence intervals (dashed lines) derived from restricted (natural) cubic spline meta-regression (*glst*) with three knots at 0, 34 and 84 increments in GL (per 80 g/d in 2000 kcal (8400 kJ) diet). For comparison a log-linear dose response is also shown (red). There was no significant evidence for departure from log linearity (*P* > 0.194 for the secondary spline determinant). Thus the plots (red compared with black) overlapped. Observations were truncated at 100g GL/d in 2000 kcal due to scarcity of increments above this value. The log-linear dose-response T2D-GL risk relation increased by 27% per 80 g GL/d in 2000 kcal diet. RR was 1.27 (1.19–1.36) (*P* < 0.001, *n* = 13). The cubic spline curve reached 1.42 (1.27–1.59) for a 100 g increment. There were 44 incremental observations from 13 studies with CORR > 0.55. The mean GL at the intercept (zero increment in GL) was 105 g/d in 2000 kcal (8400 kJ) based on the glucose (GI = 100) scale. Studies were from: Hodge et al. 2004 [39], Hopping et al. 2010 (5 studies, fCA, mCA, fJA, mJA, mNH) [44], Mekary et al. 2011 [30], Sahyoun et al. 2008 [45], Salmeron et al. 1997 in men [31], Sakurai et al. 2012 [41], Schulze et al. 2004 [33], van Woudenbergh et al. 2011 [42] and Villegas et al. 2007 [34]. Abbreviations: GL, glycemic load; RR, relative risk; T2D, type 2 diabetes.

**Table 1 nutrients-11-01436-t001:** Bradford-Hill criteria (1–9) for assessment of causation in prospective cohort studies; definitions used in the present study, and cost–benefits (Point 10).

No *^a^*	Criterion	Bradford Hill’s Definition *^a^*	Definition in This Study
(1)	Strength of association	An association between disease and exposure needs to define a strong association, which depends on the phenomenon being addressed.	An association of significant strength is defined as one with an RR < 0.83 or >1.20 *^b,c,d^* in the expected direction with statistical significance of *P* < 0.05 obtained from meta-analyses of relevant studies and least deviant 95% CL < 0.91 or >1.10 *^c^* respectively.
(2)	Consistency	Finding of an association needs to be replicated in other studies.	*i:* Consistency of association was defined as one in which ≥3 *^e^* studies assessed by meta- analysis yielded an inconsistency statistic (I^2^) that was zero or non-significant *P* > 0.05. *ii:* Ideally, associations are found in different peoples, places, times and using different assessment tools.
(3)	Specificity *^f^*	Specific exposure is related to only one disease. Bradford Hill states this criterion should not be over emphasized.	The specified association for the disease incidence is related to the exposure variable only. Potentially confounding exposures, non-dietary and dietary, are adjusted for in the original observational studies or assessed by relevant sensitivity analysis during meta-analyses *^f^*.
(4)	Temporality	Exposure must precede incidence of disease.	Study designs must be temporally correct, here achieved by restriction to observations from prospective cohort studies with absence of disease at baseline, and RCTs with surrogate endpoints and/or incident disease.
(5)	Biological gradient (dose-response)	Risk of disease is increased (or decreased) as the level of exposure increases (or decreases).	Coefficient for trend is significant (*P* < 0.05) either within original studies or following meta-regression analysis. Dose-response curves should fit curvature if evident; otherwise curvature may contribute to I^2^.
(6)	Plausibility	An association makes biological sense, which depends on current knowledge.	Mechanism(s) are known by which incident disease is expected to develop upon introduction of people to the exposure of concern.
(7)	Experiment	Evidence from RCTs, or strong support from less rigorous trials. Evidence can include increased or decreased incidence of disease or surrogate markers according to increased or decreased exposure.	Evidence from animal and human studies, as described by this criterion (see left).
(8)	Analogy	Knowledge of other effects, and exposures having similar result in one or more similar diseases.	Knowledge of other exposures having similar effects, result in similar diseases. Similar effects mean effects on surrogate markers or incident disease.
(9)	Coherence	Causality should not seriously conflict with the knowledge on natural history and biology of disease.	I. Association is supported by evidence on surrogate risk factors. II Observations are coherent with other relevant epidemiological observations. III Observations do not conflict with, but complement or enhance, food and dietary advices for disease risk reduction, unless these are falsely based.
(10)	Cost benefit *^g^*	Not a Bradford Hill criterion, but is important to realizing the financial costs or savings that may result from resultant modification of the disease burden.	The potential proportion of health care costs savable.

a. The order of Hill’s criteria here differs from those presented by Bradford Hill [5], who assigned Coherence to number 7, whereas here it is 9. b. RR values meeting these criteria are regarded as sufficiently strong for consideration in public health [14,15]. c. Whether an RR is beneficial or harmful is dependent on whether exposure rises or falls, and whether the referent cohort is placed at the highest or lowest exposure. RR values of 0.83 and 1.20 are of equivalent strength for beneficial and harmful associations respectively when the referent is at lowest category of exposure. The least deviant 95% CL values (0.91 and 1.10) likewise are of similar strength and were chosen to reflect that a change in risk of 10% has been considered to be clinically significant [16]. d. For consistency, RR values should be based on a meaningful and explicit range of exposures/doses/intakes. These ranges should be interconvertible among studies, which requires that each study should specify the units of exposure and units of association. e. Three studies are a minimum for providing an estimate of variance among studies, for accuracy more studies are desirable but in practice can be from 3 to 10 studies or more. Ten or more studies provides scope for examination of covariates, which is desirable when assessing nutritional factors [11,17], because they are not as strong as environmental toxicities (e.g., smoking). f. Characteristics of foods or diets can associate with more than one disease; therefore the original criterion was modified here, so to maintain the link between a specified disease and a specified exposure. g. Cost benefit analysis was indicated useful when applying the knowledge gained [10]. Abbreviations: CL, confidence limit; I^2^, inconsistency i.e., the ratio of the variance among studies expressed as a % of the total of variances among and within studies; P, probability; RCTs, randomized controlled (intervention) studies; RR, relative risk.

**Table 2 nutrients-11-01436-t002:** Summary of the outcomes of this review according to the Bradford-Hill criteria (Points 1–9) and potential cost benefit (Point 10) ^a^.

	Criterion/Outcome
(1)	Strength of association
	Critical meta-analyses of prospective cohort studies show both the T2D-GI and the T2D-GL risk relations are sufficiently strong (RR > 1.20, lower 95% CL > 1.10) to warrant action in favor of public health.
(2)	Consistency
	When robust approaches to data synthesis are used, the results among prospective cohort studies are sufficiently consistent both without and with adjustment for validity correlations to support a conclusion that the risks relations are of biological significance. The risks to health occur to a greater or lesser extent under different circumstances, e.g., different ethnic ancestry, places, times, foods, in addition to men, women, and higher and lower BMI sub-populations of women.
(3)	Specificity
	*Non-Dietary factors:* Considering all eligible prospective cohort studies on GI or GL together and recognizing the potential for residual confounding, major non-dietary factors were unable to explain the strength of association between T2D and GI or GL. The non-dietary factors included age, race, weight, smoking status, physical activity and family history of diabetes, as well as menopausal status and use of post-menopausal hormonal therapy in studies of women. *Dietary factors:* Similarly, intakes of total energy, *trans*-fats, saturated fats, protein, fiber, or cereal fiber and alcohol in the original prospective cohort studies do not explain the study-level strengths of association between T2D and GI or GL. T2D-GI and GL relations and T2D-fiber (or cereal fiber) relations are independent and additive. Alcohol intake may attenuate the T2D-GL risk relation, thus a sex-difference in alcohol consumption may explain a sex-difference in the T2D-GL relation. The strength of the T2D-GL risk relation found is independent of simultaneous putative confounding by simultaneous adjustments for protein, energy and fat or fats ^b^.
(4)	Temporality
	A temporal relationship of GI or GL to prevent or delay T2D is indicated by 3 independent sources of data: (1) Prospective cohort studies in which incident T2D occurs after consumption of diets different in GI or GL. (2) Randomized controlled intervention trials that show plausible mechanisms and relevant changes in T2D risk factors. (3) Randomized controlled intervention trials that use tolerable doses of carbohydrate inhibitors to slow rather than prevent carbohydrate digestion in the small intestine (thereby lowering dietary GI or GL) result in lower or delayed incidence of T2D. These inhibitors act only in the gut and are not absorbed into the circulation.
(5)	Biological gradient (dose-response)
	Highly powered prospective cohort studies and dose-response meta-analyses show the T2D-GI and the T2D-GL risk relations are dose dependent over a wide range of GI and GL.
(6)	Plausibility (mechanisms)
	At least three complementary mechanistic chain of events link diets of higher GI and GL to T2D in a causative manner. These include elevation of glucotoxicity, lipotoxicity and ponderal toxicity including central obesity. All three compromises beta-cell function. Ponderal toxicity is modest if related solely to the effects GI or GL on the rate and extent weight loss. Effects on central obesity may arise in the absence of significant body weight change. Restricting the intake of high GI foods in the context of limit carbohydrate intake may be supportive of both body weight and central obesity reduction.
(7)	Experimental evidence
	Experimental studies in animals and humans show diets of higher GI and GL cause significant features of T2D while diets of lower GI and GL show the reverse.
(8)	Analogy
	Evidence from randomized controlled trials indicate that inhibitors used to slow, carbohydrate digestion (analogous to lowering dietary GI and GL) can prevent or delay progression of impaired glucose tolerance to diabetes.
(9)	Coherence
	*Surrogate markers of T2D:* interventional evidence on surrogate markers of T2D risk and reversal of disease progression, as monitored by fasting blood glucose and glycated proteins, are coherent with robust observations on incident T2D-GI and GL risk relations. *BMI, CHD and cancer:* Both overweight and obese persons are at risk ofT2D, and both are at lower risk when consuming diets lower in GI or GL. T2D associates with CHD and colorectal cancer, and persons consuming diets of lower GI and GL are also at lower risk for both conditions. *Food and nutrition advice (from the Discussion):* Low GI and GL advice does not conflict with current healthy eating dietary advice and should be applied in the context of healthy food-and nutrient-based advice. At present, conventional dietary advice is inadequate for identifying low GI or GL foods or diets.
(10)	Cost benefit (from Discussion)
	Advice on lowering the GI and GL of diets has potential to make significant savings from national health budgets and GDP through preventive action to lower the burden of disease. The advice is consistent with sustainable development of earth systems.

a. As defined and used in this review (Table 1). b. The reported T2D-GI and GL risk relations are those falling at the average of the study population means or medians for factors above [see Specificity (3)] for which the T2D-GI and GL relations were adjusted in the original studies. Abbreviations: GDP, gross domestic product; GI, glycemic index; GL, glycemic load; BMI, body mass index; T2D, type 2 diabetes.

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
