# Peer review of "Dietary Glycemic Index and Load and the Risk of Type 2 Diabetes: Assessment of Causal Relations"

_nutrients, 2019, doi:10.3390/nu11061436_

Round 1
Reviewer 1 Report
Interesting manuscript in which the authors, from a causal perspective using the Bradford Hill criteria, perform an extensive review of the impact of low GI and GL carbohydrate diets on the incidence of type 2 diabetes.
It is based on another manuscript sent at the same time for publication, that I do not know its content, but that I understand that the acceptance of its reviewers is a fundamental precondition for the acceptance of this one.
One is not sure that a RR of 1.20 means a strong association. Although it has been used previously in a couple of studies (references 14 and 15, not 16), there is still no cutoff point defining clinically meaningful effects. Mention more clearly in the discussion section.
It seems interesting that the T2D-GI and Gl risk relation was lower in moderate alcohol drinkers. Although it is not the objective of this review I am sure that the readers would appreciate a more detailed writing of the alcohol section.
Although the manuscript is well written, it presents some typos that should be corrected. (lines 272, 281, 352, 372, 419 [RR 2.02?], 1109, 1110, 1157).
Author Response
Response to Reviewer 1
Comments and Suggestions for Authors
1) Interesting manuscript in which the authors, from a causal perspective using the Bradford Hill criteria, perform an extensive review of the impact of low GI and GL carbohydrate diets on the incidence of type 2 diabetes.
Thank you for your agreeable comments.
2) It is based on another manuscript sent at the same time for publication, that I do not know its content, but that I understand that the acceptance of its reviewers is a fundamental precondition for the acceptance of this one.
This is true.
3) One is not sure that a RR of 1.20 means a strong association. Although it has been used previously in a couple of studies (references 14 and 15, not 16), there is still no cutoff point defining clinically meaningful effects. Mention more clearly in the discussion section.
Thank you.
In Table 1 footnote c we stated:
“The least deviant 95% CL values (0.91 and 1.10) likewise are of similar strength and were chosen to reflect that a change in risk of 10% has been considered to be clinically significant [16]. “
However, we welcome the reviewers suggestion to mention this in the discussion [where it less easily escapes attention]. Hence we introduced a new 2nd paragraph in the Discussion .
4) It seems interesting that the T2D-GI and Gl risk relation was lower in moderate alcohol drinkers. Although it is not the objective of this review I am sure that the readers would appreciate a more detailed writing of the alcohol section.
Thank you for this suggestion. We partially rewrote the section on alcohol as a risk modifier at orginal lines 280 to 289, and extended it as follows:
“Alcohol. Both the T2D-GI and GL risk relations were attenuated in studies in which the mean population alcohol consumption was moderately high (>15 g/d). Likewise, a higher than average alcohol consumption completely attenuated the coronary heart disease-GI and GL risk relations [14]. This attenuation may be model dependent and for T2D (and CHD [14]) was found when the validity correlation was also a covariate. This attenuation was first identified within a prospective cohort study for GL though not for GI [30]. The observations within this [30] study were weaker than we found among studies for T2D [1] and CHD [14], which might be due to the within-study observations applying to women only whereas the among-study observations were in men and women combined. A possibility exists that a sex difference in the size of the T2D-GI and GL risk relations may be due to a sex difference in alcohol consumption. [1] Attenuation by alcohol consumption may explain why observations from the Healthy Professionals Follow-up Study [19] and 3 other studies [31, 32, 33] showed lower than expected associations between T2D and GL. Epidemiological observations do not identify a mechanism of action, which could be many [30]. Nevertheless, our evidence points toward higher T2D (and CHD [14])-GI and GL risk relations for alcohol abstainers than indicated by the primary outcome risk relations.”
5) Although the manuscript is well written, it presents some typos that should be corrected. (lines 272, 281, 352, 372, 419 [RR 2.02?], 1109, 1110, 1157).
Corrections were made at all these lines except 1109 and 1110 among the references where no typo was evident.

Reviewer 2 Report
This is a very comprehensive study of the effects of GI and GL. A fine paper.
There are several spelling mistakes lines 270-282. Other than this I have no comments
Author Response
Reviewer 2
Comments and Suggestions for Authors
This is a very comprehensive study of the effects of GI and GL. A fine paper.
Thank you for your favourable comment.
There are several spelling mistakes lines 270-282. Other than this I have no comments
Thank you. Corrections have been made.
1. Line 130 - what is EQM?..... In the same line "of" is "to"
Thank you. Extreme Quantile Meta-analysis ( giving the risk ratio, i.e risk at the highest quantile for exposure / risk at the lowest quantile of exposure)
2. Line 158 - validity correlations - explain this the first time used.
Thank your we inserted a footnote: “The validity correlation is the correlation between the intakes of a nutrient or type of food estimated from a dietary instrument (usually a food frequency questionnaire) and the intake determined using food records among a sample of the population to be studied. Optimally these correlations are energy adjusted and deattenuated [25] and are sufficiently large to validate the questionnaire when judged against a predefined value [26].
3. Lines 205-209 - run on sentence. Hard to understand as written.
Rewrite please.
Thank you, we simplified the text to:
Note that the criteria of interest for public health in Table 1 (Strength) make no specific requirement about the level of inconsistency (I2) of results among studies. This is because meeting the criteria would not be possible if inconsistency was too high. This is useful because I2 is an imprecise measure, [27, 28]. Indeed, I2 can be zero when studies are imprecise and can be high when a risk relation is large.
4. Line 267 - what is FDH?
This typo was supposed to be FHD
It was replaced with “family history of diabetes”
5. Line 272 – versus
“versus” was found only in the two Figure legends. We replace it with ‘compared with”
6. Independently is an adverb; I believe independent is correct......Line 275 -
Thank you this problem was resolved at point 6
7. Line 295 - follow UP
Indeed ‘follow up’
8. line 304 - is it "Over all" or is it "Overall"
“Over all” in this sentence.
9. Lines 352, 373 - spelling
Only one spelling error was found in the original submission: “resultsed” was replace with ‘resulted’
10. Line 462 - FUY? What is that?
“FUY (number of follow-up years)” now stated.
11. Lines 470 - run on sentence. Divide into 2 sentences for easier reading.
Thank you, done; this was helpful.
12. Line 479 - you never mentioned menopause until now.
This was mentioned at line 362 to363 in the original submission “….menopausal status in women and related hormone use plus oral contraceptive use (n=2) (Table S17 in [1]).
13. Line 498 - here you use the word "causative". This seems out of place since you are trying to establish causality throughout this paper. What you have is a strong association
Yes good point. We only need to say temporal relations
14. Line 582 - alcohol does not appear in this section (it was discussed before). The statement is out of place.
Removed from this section criteria
15. Line 647 - spelling
Protei was reunited with n
16. Line 831 - what does being sedentary have to do with this sentence?
We deleted the parenthetical note that included ‘sedentary’ , because it was not essential.
17. Lines 838-846 - hard to follow. Could you simplify or make clearer
This was simplified in response to comments from Co-authors.
18. Line 865 - was the fact brought here mentioned before when you were discussing fiber as a separate category?
The question was unclear. The discussion looked find to the corresponding author and no other co-author has mentions anything problematic about the content of the text at this point..

Reviewer 3 Report
This MS addresses the role of dietary glycemic index (GI) and glycemic load (GL) the association with risk of T2D and the extent to which causality can be inferred. The approach is to interpret the authors’ recent meta-analyses and related work in the literature with respect to adherence to Bradford-Hill’s criteria. The conclusion is that “all nine of the Hill’s criteria were met for GI and GL indicating that we can be confident of a role for GI and GL as causal factors contributing to incident T2D.”
I do not share the authors’ confidence. I have numerous criticisms, particularly to the application of Bradford-Hill’s criteria which are violated in spirit and substance in the MS. Under usual circumstances, I would suggest rejecting this paper but the authors are well-known and presumed authorities and the GI/Gl idea is widely considered if not universally accepted and it is valuable to have the major proponents make their case. I would suggest that the MS be published but it must be recognized that the methods and conclusions are controversial. Therefore, I think that it should be published, if and only if, I and my colleagues can write a rebuttal that would be published alongside the original.
My objections: GI/GL continue to be somewhat problematical primarily because of the questionable idea that the response to a food in one context can carry over into all cases for all people, particularly all people with different immediate and extended dietary histories. In addition, I would strongly agree with the comment in the Abstract “the causal role of carbohydrate nutrition remains controversial.” In my view, the major part of that controversy rests with the role of total carbohydrate in diabetes where we have the most hard data at least on treatment of diabetes and for which GI/GL maintains an unpredictable relationship. The nearly total absence of discussion of low-carbohydrate diets is indicative of the bias in the MS.
While application of Hill’s criteria is certainly appropriate, the presentation in the MS seems opposite in both the spirit and details to what Hill had in mind (MS ref. 6). Hill was very clear that his criteria did not represent cut-and-dried rules but rather followed from common sense. Hill emphasized that there was nothing in statistics that could answer the question of causality. In this, he might have anticipated the reaction to The Cult of Statistical Significance as in the title of a book critical of misapplication of statistics.
The major fault — in my view, fatal — lies with the definition of what Hill considered his major criterion, the strength of the association. The MS correctly states that “the association between exposure and disease needs to be strong.” However, the definition for “this study is defined as one with an RR<0.83 or="">1.20b.c.d in the expected direction with statistical significance of P<0.05 obtained from meta-analyses of relevant studies and least deviant 95% CL <0.91 or="">1.10 c respectively.” While such poor RR might be acceptable in a prospective drug study where it is clear who got the drug and who didn’t, it is not reasonable in a dietary study where there is a multicomponent independent variable. Hill’s own studies are widely cited for having RR of 20 for smoking and lung cancer or 30 for heavy smoking. He was, in fact, not sure that an RR of 2 would imply causality in smoking and heart disease. It is also commonly pointed out that an RR of 2 is generally a minimum in introducing epidemiological data into evidence in a toxic tort case in a court of law.
The reference b cited as superscript in the passage above are:
Livesey, G., et al. (MS ref. 14) which says, in agreement with the MS, that it “is important for public health when RR is greater than 1.20 and its lower confidence limit is greater than 1.10 from the 10th to 90th percentile of nutrient intake…” citing “(Livesey,, et al, unpublished data, 2018). This criterion was met for the CHD-carbohydrate RR for high GI carbohydrate….” and,
Mente, et al. (ref. 15) with similar RR of 1.2 . This reference admitted that they “may be criticized for creating arbitrary definitions of strong, moderate, and weak evidence, although these classifications have face validity and similar scoring systems have been used to assess the evidence of causation from observational studies.53,54 Second, we had to derive the RR cutoff points to define a strong association from the distribution of RR values in cohort studies because the true cutoff points for defining clinically meaningful effects are not known.” This seems seems to imply that usage is a criterion for validity.
Finally, the conclusion “Potentially, approx. 23% of these costs may relate to the consumption of diets too high in GI or GL when considering an RR of ~1.3….this would suggest ~46% of diabetes costs may be associated with high GI and GL diets in countries with diets particularly high in GI or GL” is a strong statement that is entirely speculative.
Author Response
Review 3
Comments and Suggestions for Authors
This MS addresses the role of dietary glycemic index (GI) and glycemic load (GL) the association with risk of T2D and the extent to which causality can be inferred. The approach is to interpret the authors’ recent meta-analyses and related work in the literature with respect to adherence to Bradford-Hill’s criteria. The conclusion is that “all nine of the Hill’s criteria were met for GI and GL indicating that we can be confident of a role for GI and GL as causal factors contributing to incident T2D.”
Thank you for recognising the authority of the authors.
I do not share the authors’ confidence. I have numerous criticisms, particularly to the application of Bradford-Hill’s criteria which are violated in spirit and substance in the MS. Under usual circumstances, I would suggest rejecting this paper but the authors are well-known and presumed authorities and the GI/Gl idea is widely considered if not universally accepted and it is valuable to have the major proponents make their case. I would suggest that the MS be published but it must be recognized that the methods and conclusions are controversial. Therefore, I think that it should be published, if and only if, I and my colleagues can write a rebuttal that would be published alongside the original.
Thank you for recommending publication.
My objections: GI/GL continue to be somewhat problematical primarily because of the questionable idea that the response to a food in one context can carry over into all cases for all people, particularly all people with different immediate and extended dietary histories. In addition, I would strongly agree with the comment in the Abstract “the causal role of carbohydrate nutrition remains controversial.” In my view, the major part of that controversy rests with the role of totalcarbohydrate in diabetes where we have the most hard data at least on treatment of diabetes and for which GI/GL maintains an unpredictable relationship. The nearly total absence of discussion of low-carbohydrate diets is indicative of the bias in the MS.
The reviewer states “Objection …because of the questionable idea that the response to a food in one context can carry over into all cases for all people, particularly all people with different immediate and extended dietary histories”. There are four issues here;
1. We certain don’t claim this.
2. National dietary guidelines usually apply to people in general, specifically for maintenance of growth and health at each age stage.
3. The objection would apply to all nutrients in national nutrition guideline; therefore we feel the objection is misplaced.
4. The comment about application to all cases for all people is taking an extreme position because we know there is some variability in response to almost everything in our diets and other exposures, but that doesn’t stop us from making general recommendations or policies.
5. We already make clear that the observational results apply to persons of European ancestry and East Asian ancestry, while it is unknown whether they apply to persons of South American ancestry or African ancestry in the co-submitted manuscript.
6 In the present manuscript we suspect the reviewer misunderstands the application of Hills criteria, which suggests finding the observation in question under different circumstance (not all circumstances). In respect of this we add a short paragraph immediately before the criterion conclusion under the Section “Consistency of association” which reads:
“It is important to realize the Bradford Hill suggests making the same (similar) observations under “different” circumstances, not ‘all’ circumstances. Here for example, observations on the T2D-GI and GL risk relations met the criteria for public health interest in persons of European ancestry and East Asian ancestry. There was insufficient information on the strength of these relations in persons of South American ancestry and African ancestry, but as Hill would say, this is not proof of no T2D-GI or GL risk relation but would support the case for causation. ”
Thank you, in the abstract we overgeneralised by saying “the causal role of carbohydrate in nutrition”. Therefore we amended this to “the causal role of carbohydrate quality in nutrition”
Thank you again, we agree more information has become available on carbohydrate amount in the treatment of diabetes. Clearly our concern throughout the paper has been with carbohydrate quality, GI and GL specifically. GI and GL have particular relevance to carbohydrate amount in nutritional therapy for diabetes, in which we have had co-author involvement in the UK and through the Royal College of General practitioners, with tremendous success (See Unwin, J. D.; Haslam, D.; Livesey, G., It is the glycaemic response to, not the carbohydrate content of food that matters in diabetes and obesity: The glycaemic index revisited. Journal of Insulin Resistance 2016, 1 (a8), 1-9), which was already cited in the paper.
While application of Hill’s criteria is certainly appropriate, the presentation in the MS seems opposite in both the spirit and details to what Hill had in mind (MS ref. 6). Hill was very clear that his criteria did not represent cut-and-dried rules but rather followed from common sense. Hill emphasized that there was nothing in statistics that could answer the question of causality. In this, he might have anticipated the reaction to The Cult of Statistical Significance as in the title of a book critical of misapplication of statistics.
Thank you for this reminder. It was precisely because of this that Hill communicated Hill’s criteria and why we used them.
The major fault — in my view, fatal — lies with the definition of what Hill considered his major criterion, the strength of the association. The MS correctly states that “the association between exposure and disease needs to be strong.” However, the definition for “this study is defined as one with an RR<0.83 or="">1.20b.c.d in the expected direction with statistical significance of P<0.05 obtained from meta-analyses of relevant studies and least deviant 95% CL <0.91 or="">1.10 c respectively.” While such poor RR might be acceptable in a prospective drug study where it is clear who got the drug and who didn’t, it is not reasonable in a dietary study where there is a multicomponent independent variable. Hill’s own studies are widely cited for having RR of 20 for smoking and lung cancer or 30 for heavy smoking. He was, in fact, not sure that an RR of 2 would imply causality in smoking and heart disease. It is also commonly pointed out that an RR of 2 is generally a minimum in introducing epidemiological data into evidence in a toxic tort case in a court of law.
Thank you for placing your point of view. There is ample literature to show that this view is not accepted in nutritional epidemiology:
1) A wide range of nutrients, foods and diets (33) have relative risks that all with the range 0.62 to 1.32. [1]
2) Dietary guidelines committees worldwide have support recommendations on the basis of such results, often before information from RCT is available [2], Meanwhile prospective cohort studies can be every bit as good as RCTs although it is common to downgrade them. Intervention studies are however are also subject to confounding [3]
3) No other field of nutritional science can provide direct information on relations between nutrition and health in free-living human populations [4]
3) One thing that Bradford Hill makes clear is that no evidence sufficient to warrant action comes without its critics.
4) A change of 50% in risk (e.g, a relative risk of 1.5) may be considered strong
in nutritional epidemiology, so the commonly cited criterion of relative risks<2.0 being suspect does not necessarily apply to nutritional epidemiology” [4]. A 20% increase or decrease in risk related to a food or nutrient could be both biologically plausible and have a meaningful effect on public health as a result of wide exposure (everyone is exposed to nutrients), provided adequate control for confounding factors [4]
5. It is commonly acceptable to use epidemiological results in development of nutrition guidelines [4], irrespective of low relatives risks by comparison with toxic substances to which one is expose to by tobacco smoking. Why is this?
· Firstly epidemiological methodology has improves over the last 50 years since Bradford Hills landmark paper [2].
· Second, statistical methodology and knowhow for combining study results has improved over the last 50 years too (although too easily used by persons without adequate nutrition knowledge).
· Third, the results are highly important because everyone eats and has to eat, there is no escape but death, hence even small risk relations have large absolute risks for whole populations.
· Nutritional epidemiology has a past a present and a future in nutrition guideline development [5].
The reference b cited as superscript in the passage above are:
Livesey, G., et al. (MS ref. 14) which says, in agreement with the MS, that it “is important for public health when RR is greater than 1.20 and its lower confidence limit is greater than 1.10 from the 10th to 90th percentile of nutrient intake…” citing “(Livesey,, et al, unpublished data, 2018). This criterion was met for the CHD-carbohydrate RR for high GI carbohydrate….” and,
Thank you. We introduced a new 2nd paragraph in the Discussion section.
Mente, et al. (ref. 15) with similar RR of 1.2 . This reference admitted that they “may be criticized for creating arbitrary definitions of strong, moderate, and weak evidence, although these classifications have face validity and similar scoring systems have been used to assess the evidence of causation from observational studies.53,54 Second, we had to derive the RR cutoff points to define a strong association from the distribution of RR values in cohort studies because the true cutoff points for defining clinically meaningful effects are not known.” This seems seems to imply that usage is a criterion for validity.
Thank you. We address this also in the new 2nd paragraph in the Discussion section.
Finally, the conclusion “Potentially, approx. 23% of these costs may relate to the consumption of diets too high in GI or GL when considering an RR of ~1.3….this would suggest ~46% of diabetes costs may be associated with high GI and GL diets in countries with diets particularly high in GI or GL” is a strong statement that is entirely speculative.
Thank you. We agree we made a strong statement. We also agree that it is speculative because it is theoretical or hypothetical as are all projections of this type in advance of application. However, we recognise that a full cost benefit was not conducted and therefore deleted the matter concerning “….this would suggest ~46% of diabetes costs may be associated with high GI and GL diets in countries with diets particularly high in GI or GL” and matter related to “GDP”.
References
1. Mente, A.; de Koning, L.; Shannon, H. S.; Anand, S. S., A systematic review of the evidence supporting a causal link between dietary factors and coronary heart disease. Arch Intern Med 2009, 169, 659-669.
2. Satija, A.; Yu, E.; Willett, W. C.; Hu, F. B., Understanding nutritional epidemiology and its role in policy. Adv Nutr 2015, 6, 5-18.
3. Livesey, G.; Taylor, R.; Hulshof, T.; Howlett, J., Glycemic response and health a systematic review and meta-analysis: relations between dietary glycemic properties and health outcomes. Am J Clin Nutr. 2008, 87, 258S-268S.
4. Byers, T.; Lyle, B., The role of epidemiology in determining when evidence is sufficient to support nutrition recommendations. Summary statement. Am J Clin Nutr 1999, 69, 1365S-1367S.
5. Byers, T., The role of epidemiology in developing nutritional recommendations: past, present, and future. Am J Clin Nutr 1999, 69, 1304S-1308S.

Round 2
Reviewer 3 Report
NOTE: 3 figures in attached file.
This MS addresses the role of dietary glycemic index and glycemic load (GI/GL) and the association with risk of T2D and the extent to which causality can be inferred. The approach is to interpret the authors’ recent meta-analyses and related work in the literature with respect to adherence to Bradford-Hill’s criteria. The conclusion is that “all nine of the Hill’s criteria were met for GI and GL indicating that we can be confident of a role for GI and GL as causal factors contributing to incident T2D.”
I have previously submitted some comments which the authors have answered in part. My objections remain and I still believe the MS is completely opposite in spirit and substance to Bradford Hill’s criteria. In particular, the authors simply redefine Hill's criteria for large effects. The MS is based largely on opinion and conclusions, rather than real data. In addition, the major limitations of GI/GL are underplayed and, most of all, the MS violates minimal standards of scientific discourse by ignoring the effect of total carbohydrate effect, which is the major alternative to GI/GL and of which GL is a subset. As suggested previously, the issues should receive the open discussion they deserve. Therefore, as before, I recommend publication under the condition that I and my colleagues write a rebuttal to be published alongside the original. There are numerous precedents and I think it will serve Nutrient’s readers and the general public well. My objections:
Hill’s Major criterion is violated.
Hill was very clear that his criteria did not represent cut-and-dried rules but rather followed from common sense and experience. Hill’s first criterion was the magnitude of the association and this is seen in the context of the data. I reiterated in my previous comments the oft-quoted values of an RR of 20 for smoking and lung cancer or 30 for heavy smoking and the general idea that an RR of 2 was a kind of default value assuming reasonable error in the independent variable. In response to my original comment , the authors answers were:
Thank you for placing your point of view. There is ample literature to show that this view is not accepted in nutritional epidemiology:
The question is why not? Many researchers cannot understand how the multitude of nutritional epidemiology papers can be published with HR in the range of 1.3 to 1.5 or even as low as 1.12, especially given that most, have, in fact, universally accepted high error in food consumed based on FFQ. The authors provide 2 references by Byers. An excerpt from one of these:
“The Committee on Diet and Health of the National Research… adapted the Hill (7) criteria for causality to interpret the diverse nutritional literature, qualitatively judging whether the totality of evidence pointed to an association that was strong, showed a dose-response relation,… Even the committee recognized, however, that these criteria are of limited use in nutritional epidemiology. Not all meaningful associations are expected to be strong (hence a set of studies will not necessarily show either a dose-response relation or consistency), nutritional factors are not specific because they may affect several diseases similarly, and biologic plausibility is a product of the state of knowledge at any given time and subjective imagination.
The Hill criteria for causality are therefore of limited practical utility in nutritional epidemiology….” — Byers, (1999) Am J Clin Nutr;69(suppl):1304S–8S.
There is adherence to Hill’s criteria or not? The conclusion is that nutritional epidemiology can ignore Hill’s criteria without suggesting alternatives.
A wide range of nutrients, foods and diets (33) have relative risks that all with the range 0.62 to 1.32. [1]
Most of these do not conform to Hill’s criteria and their persistence can not change the criteria. It is likely that most do not address the criteria at all, or provide any alternative. Most people think that 57:43 is too close to even odds to be taken seriously.
GL/GI is not reliable
My previous comment and response:
The reviewer states “Objection …because of the questionable idea that the response to a food in one context can carry over into all cases for all people, particularly all people with different immediate and extended dietary histories”. There are four issues here;
1. We certain don’t claim this.
I was not clear on this. What I meant was that GI/GL is determined from averages of measurements on a particular population from the glucose AUC for fixed time interval. The assumption is that the number will be relevant to other patients and populations without consideration of their individual history, that is, it is assumed, as far as I know that GI/GL is a characteristic of the food. This seems unreasonable and, in fact, measurements of GI/GL have high variability. Since they are derived from phenomenology without any mechanism, there is no basis for considering a particular distribution or whether averages are appropriate. Figure 1 (attached) from Wolever, et al. (2003) European Journal of Clinical Nutrition (2003) 57, 475–482 where individual points are themselves averages seems telling:
“Carbohydrate Quality” is not defined for GI/GL
It is not obvious what “carbohydrate quality” means. GI/GL is defined by the glucose response and GL is a subset of total carbohydrate. The implication that the glucose response is somehow more biologically meaningful than the total effect of carbohydrate per se is not supported. The rationale for focussing on total carbohydrate is not just the change in blood glucose but the effect on hormones and other metabolites. In an extreme case, foods with similar GI/GL may have different incretin affects and different changes in incretin n diabetes or other conditions. see Figure 2 from Jutonen, K. (2002) AJCN 75, 254-265
GI/GL is not the best predictor of dietary practice.
Figure 3 Feinman, et al. (2015) Nutrition 31, 1-14
“Carbohydrate restriction is the single most effective intervention for reducing all of the features of metabolic syndrome. Figure 9 shows the results from a study comparing a low glycemic index (low-GI diet) with a standard high-cereal diet in 210 people with type 2 diabetes [70].… The difference…. stands as the single most telling indication of the potential for carbohydrate restriction in diabetes. The low-carbohydrate diet (reddish bar) shows the greatest decrease in TG, as well as decrease in weight, HbA1c and glucose and a greater increase in HDL.”
While this represents the effects on treatment, the relevance to prevention is at least as plausible as deductions from epidemiology.

Author Response
The following document is the reply to the reviewer's comments.
